# DNA methylation profiling identifies TBKBP1 as potent amplifier of cytotoxic activity in CMV-specific human CD8+ T cells

Zheng Yu[1]☯, Varun Sasidharan-Nair[1]☯, Thalea Buchta[2,3], Agnes Bonifacius[4,5], Fawad Khan[6,7], Beate Pietzsch[1], Hosein Ahmadi[1], Michael Beckstette[1]¤, Jana Niemz[1], Philipp Hilgendorf[8], Philip Mausberg[4], Andreas Keller[9,10], Christine Falk[5,11], Dirk H. Busch[12,13], Kilian Schober[8,14], Luka Cicin-Sain[5,6,7], Fabian Müller[15], Melanie M. Brinkmann[2,3], Britta Eiz-Vesper[4,5‡], Stefan Floess[1‡], Jochen Huehn🄳[1,16‡*]

1 Department Experimental Immunology, Helmholtz Centre for Infection Research, Braunschweig, Germany, 2 Institute of Genetics, Technische Universität Braunschweig, Braunschweig, Germany, 3 Research Group Virology and Innate Immunity, Helmholtz Centre for Infection Research, Braunschweig, Germany, 4 Institute of Transfusion Medicine and Transplant Engineering, Hannover Medical School, Hannover, Germany, 5 German Center for Infection Research (DZIF), Thematical Translation Unit-Immunocompromised Host (TTU-IICH), partner site Hannover-Braunschweig, Germany, 6 Department Viral Immunology, Helmholtz Centre for Infection Research, Braunschweig, Germany, 7 Centre for Individualized Infection Medicine (CIIM), a joint venture of HZI and Hannover Medical School, Hannover, Germany, 8 Mikrobiologisches Institut–Klinische Mikrobiologie, Immunologie und Hygiene, Universitätsklinikum Erlangen und Friedrich-Alexander-Universität (FAU) Erlangen-Nürnberg, Erlangen, Germany, 9 Clinical Bioinformatics, Saarland University, Saarbrücken, Germany, 10 Helmholtz Institute for Pharmaceutical Research Saarland (HIPS)-Helmholtz Centre for Infection Research (HZI), Saarland University, Saarbrücken, Germany, 11 Institute of Transplant Immunology, Hannover Medical School, Hannover, Germany, 12 Institute for Medical Microbiology, Immunology and Hygiene, Technical University Munich (TUM), Munich, Germany, 13 German Center for Infection Research (DZIF), Thematical Translation Unit-Immunocompromised Host (TTU-IICH), partner site Munich, Germany, 14 FAU Profile Center Immunomedicine, Friedrich-Alexander-Universität (FAU) Erlangen-Nürnberg, Erlangen, Germany, 15 Integrative Cellular Biology and Bioinformatics, Saarland University, Saarbrücken, Germany, 16 Cluster of Excellence Resolving Infection Susceptibility (RESIST; EXC 2155), Hannover Medical School, Hannover, Germany

☯ These authors contributed equally to this work.
¤ Current address: Bielefeld Institute for Bioinformatics Infrastructure (BIBI), Bielefeld University, Bielefeld, Germany
‡ BE-V, SF and JH also contributed equally to this work.
* jochen.huehn@helmholtz-hzi.de

**Data Availability Statement:** Sequencing data and methylation levels reported in this paper were

## Abstract

Epigenetic mechanisms stabilize gene expression patterns during CD8+ T cell differentiation. Although adoptive transfer of virus-specific T cells is clinically applied to reduce the risk of virus infection or reactivation in immunocompromised individuals, the DNA methylation pattern of virus-specific CD8+ T cells is largely unknown. Hence, we here performed whole-genome bisulfite sequencing of cytomegalovirus-specific human CD8+ T cells and found that they display a unique DNA methylation pattern consisting of 79 differentially methylated regions (DMRs) when compared to memory CD8+ T cells. Among the top demethylated DMRs in cytomegalovirus-specific CD8+ T cells was TBKBP1, coding for TBK-binding protein 1 that can interact with TANK-binding kinase 1 (TBK1) and mediate pro-inflammatory responses in innate immune cells downstream of intracellular virus sensing. Since TBKBP1 has not yet been reported in T cells, we aimed to unravel its role in virus-specific CD8+ T

uploaded to GEO under accession number GSE245832.

**Funding:** This work was supported by the Life Science Foundation (stipend to ZY; www.life-science-stiftung.de), the Friends of the HZI foundation (stipend to VS; www.helmholtz-hzi.de/en/the-hzi/friends-of-the-hzi/activities/), the Helmholtz Association (W2/W3-090 to MMB; www.helmholtz.de), the German Research Foundation (as part of the Research Unit 2830 "Advanced Concepts in Cellular Immune Control of Cytomegalovirus", grant no 431451204 to BEV; www.dfg.de) and under Germany's Excellence Strategy (EXC 2155, grant no 390874280 to JH; www.dfg.de), and the German Federal Ministry of Education and Science (BMBF; project 01KI2013 to KS and PH; www.bmbf.de). The funders had no role in study design, data collection and analysis, decision to publish, or preparation of the manuscript.

**Competing interests:** The authors have declared that no competing interests exist.

cells. *TBKBP1* demethylation in terminal effector CD8+ T cells correlated with higher *TBKBP1* expression at both mRNA and protein level, independent of alternative splicing of *TBKBP1* transcripts. Notably, the distinct DNA methylation patterns in CD8+ T cell subsets was stable upon long-term *in vitro* culture. TBKBP1 overexpression resulted in enhanced TBK1 phosphorylation upon stimulation of CD8+ T cells and significantly improved their virus neutralization capacity. Collectively, our data demonstrate that TBKBP1 modulates virus-specific CD8+ T cell responses and could be exploited as therapeutic target to improve adoptive T cell therapies.

## Author summary

Human cytomegalovirus (CMV) is a herpesvirus that infects a significant portion of the global population. While it usually causes asymptomatic or mild infections, CMV can have severe consequences for individuals with a weakened immune system, such as stem cell or organ transplant recipients or individuals with HIV/AIDS. The immune response to CMV is characterized by expansion of virus-specific CD8+ T cells, which recognize and eliminate CMV-infected cells, thereby controlling the infection. Studies have shown that the pathogen-induced differentiation of naive to effector memory CD8+ T cells is accompanied by epigenetic changes. In order to identify the major genes involved in the functionality of CMV-specific CD8+ T cells, which are also regulated by DNA methylation, we compared the methylation profiles of CMV-specific CD8+ T cells with memory CD8+ T cells. As a result of this, we found that TBK-binding protein 1 (TBKBP1) plays a crucial role in the function of CMV-specific CD8+ T cells and enhances virus neutralization capacity upon overexpression, which has not been reported previously. This study opens the possibility of not only identifying unknown genes that contribute to the functionality of CMV-specific CD8+ T cells, but also potentially lead to improvements in adoptive T cell therapies.

## Introduction

The human cytomegalovirus (CMV) has a profound impact on the innate and adaptive immune system of the host during the three main infection phases: initial replication, persistence, and latency/reactivation [1]. A significant risk of non-relapse mortality has been reported in association with CMV reactivation after allogeneic hematopoietic stem cell transplantation (HSCT) [2,3]. Studies have shown that post-transplant CMV-infected patients have a low incidence of antiviral T cells and a delayed generation of virus-responsive T cells, suggesting that functional antiviral T cells are essential for eliminating viral infections [4,5]. During the initial period following HSCT, patients suffering from CMV reactivation exhibit a rapid reconstitution of their CD8+ T cell populations due to clonal expansion of CMV-specific T cells, leading to an unfavourable CD4:CD8 ratio [6–8]. Adoptive transfer of CMV-specific T cells has been shown to significantly reduce the risk of infection and reactivation of CMV in transplant recipients [9–13]. However, CMV-specific CD8+ T [T(CMV)] cells show a high degree of heterogeneity and possess distinct immunological functions [14–16]. Consequently, current adoptive transfer treatment strategies can still be improved to ensure optimal CD8+ T cell-mediated immunity.

Upon virus infection, naive CD8$^+$ T cells are primed with viral antigen, undergo clonal expansion and differentiate into long-lived memory or terminally differentiated effector T cells [17,18]. During this differentiation process, CD8$^+$ T cells acquire cytolytic functions that involve the action of multiple transcription factors like T-bet and Eomesodermin (EOMES), which co-operate to induce the expression of various effector molecules, including interferon gamma (IFN-γ), perforin, and granzyme B [19,20]. These effector CD8$^+$ T cells then respond to their targets directly through antigen-specific cytotoxic activity and secretion of multiple cytokines and effector molecules. The human CD8$^+$ T cell population consists of various functionally distinct subsets that can be distinguished through the changes in surface expression of homing markers and co-stimulatory molecules including CCR7, CD62L, CD27, CD28, and CD45RA during the differentiation process [21,22]. Consequently, these markers have been utilized to distinguish the phenotype of different human CD8$^+$ T cell subsets, namely naive (T$_N$), stem cell memory (T$_{SCM}$), central memory (T$_{CM}$), effector memory (T$_{EM}$), and terminal effector memory T cells re-expressing CD45RA (T$_{EMRA}$).

Accumulating evidence suggests that various epigenetic processes contribute to the T cell fate specification upon CD8$^+$ T cell differentiation [23–25]. Changes in DNA methylation patterns and histone marks were shown to play a significant role in altering the transcriptional programme of effector- or stemness-related genes during differentiation of CD8$^+$ T cell subsets [23,24,26]. Genome-wide profiling studies of naive and memory CD8$^+$ T cell subsets revealed a dynamic distribution of histone marks, including H3K4me2, H3K4me3, and H3K27me3 [27–29]. In accordance with the assumption that DNA methylation in promoter regions is typically correlated with transcriptional inactivation [30,31], several studies have demonstrated that during CD8$^+$ T cell differentiation a substantial loss of methylation marks in the promoter regions is followed by transcriptional activation of corresponding genes [32–34]. Although DNA methylation studies typically focus on promoter regions, the methylation processes observed in other genomic regions, including CpG islands/clusters (CGI), non-coding intergenic regions, and gene bodies also affect the transcription network [31]. Consequently, it is imperative to extend DNA methylation studies beyond classical promoter and transcriptional start site-specific analyses to gain a deeper understanding of the impact of epigenetic processes on T cell fate specification. Apparently, DNA methylation processes at distal regulatory domains, such as enhancers and CGIs, are often found to be negatively associated with both gene transcription and active histone modifications during cellular differentiation and phenotype commitment [35–37].

The current study focuses on the epigenetic characterisation of T(CMV) cells with respect to genome-wide DNA methylation patterns. We found that T(CMV) cells display a unique DNA methylation pattern consisting of 79 differentially methylated regions (DMRs) when compared to memory CD8$^+$ T (T$_{mem}$) cells. Among these epigenetic changes, we identified a DMR in the *TBKBP1* gene, which was further studied since its role for the cytotoxicity of CD8$^+$ T cells is unknown. Intriguingly, *TBKBP1* was found to be highly expressed in T$_{EMRA}$ cells at both mRNA and protein level, and its mRNA expression significantly correlated with the degree of demethylation. Although the *TBKBP1* DMR is closely located to an exon/intron junction, differences in DNA methylation did not result in distinct alternative splicing of the *TBKBP1* transcripts in CD8$^+$ T cell subsets. Notably, the DNA methylation patterns of the *TBKBP1* DMR in T$_N$ and T$_{EMRA}$ cells remained stable upon long-term *in vitro* culture. Retroviral overexpression of TBKBP1 in CD8$^+$ T cells resulted in TANK-binding kinase 1 (TBK1) activation, as evidenced by its increased phosphorylation status. Furthermore, CMV-specific T cells overexpressing TBKBP1 showed enhanced cytotoxicity against CMV-infected target cells along with augmented pro-inflammatory cytokine production. In summary, our results suggest that targeted demethylation of the *TBKBP1* DMR or overexpression of TBKBP1 in CD8$^+$

T cells might improve the current adoptive T cell therapy for the prevention of CMV infection and relapse.

## Results

### T(CMV) cells exhibit a unique epigenetic signature

The differentiation of naive murine CD8+ T cells into the different types of memory and effector T cells is associated with specific changes at the transcriptional and epigenetic level [23,24,29,38,39]. Studies on human CD8+ T cell memory formation similarly identified changes in DNA methylation patterns in the cytotoxicity-related genes *IFNG*, *GZMB*, and *PRF1* [40,41]. However, the global changes in DNA methylation patterns during the pathogen-induced formation of antigen-specific memory CD8+ T cells are only incompletely understood. Therefore, we isolated T(CMV) cells from 5 different CMV-seropositive donors to unravel their specific DNA methylation patterns. T(CMV) cells were identified as IFN-γ-secreting cells after restimulation with an overlapping peptide pool of the CMV-encoded pp65 protein (phosphoprotein of 65 kDa), which is the main component of the tegument layer of viral particles and an immunodominant target of T cell responses to CMV [42] (**S1A Fig**). An initial flow cytometric characterisation revealed that the vast majority of these IFN-γ+ T(CMV) cells showed a CD45RA+CD62L− $T_{EMRA}$ cell phenotype (**S1B Fig**), which is in line with recently published data [43,44]. For whole-genome bisulfite sequencing (WGBS), T(CMV) cells were isolated by flow cytometry-based cell sorting as CD3+CD4−CD8+IFN-γ+ cells, while memory CD8+ T cells ($T_{mem}$; CD3+CD4−CD8+CCR7−CD28highCD27+CD45RA−; n = 5) and naive CD8+ T cells ($T_N$; CD3+CD4−CD8+CCR7+CD28intKLRG1−CX3CR1−CD45RAhigh; n = 4) from the same donors served as controls (**S2 Fig**). For each sorted sample, genomic DNA was isolated and the genome-wide DNA methylation profile was determined by WGBS. Global analysis of the DNA methylation data revealed a strong progressive loss of DNA methylation from $T_N$ to both $T_{mem}$ and T(CMV) cells (**Fig 1A**), which is congruent with previously published data on human CD4+ memory T cells [45]. Next, we identified DMRs in pairwise comparisons of all T cell subsets. We found the highest numbers of DMRs between $T_{mem}$ vs. $T_N$ cells (11,357) and T(CMV) vs. $T_N$ cells (13,011), while a rather low number of only 104 DMRs were identified in the comparison of T(CMV) vs. $T_{mem}$ cells (**S1 Table**). Accordingly, a principal component analysis (PCA) based on genome-wide DNA methylation levels revealed a close sample group relationship between T(CMV) and $T_{mem}$ cells, whereas $T_N$ cells were placed separately along the main principal component 1 (PC1) (**Fig 1B**). This finding was confirmed by hierarchical clustering of individual samples, which further demonstrated that the majority of DMRs showed a consistent methylation status in T(CMV) and $T_{mem}$ cells, while being distinct to the largely methylated state of most DMRs in $T_N$ cells (**Fig 1C**). Yet, it is important to note that all T(CMV) and $T_{mem}$ cell samples clustered group-wise, indicating distinct methylation signatures between these cell types. The majority of DMRs across all pair-wise population comparisons were located in gene bodies and intergenic regions, while a smaller fraction was mapped to promoters (**Fig 1D**). In line with previous findings [33,46], we observed that many genes involved in the CD8+ T cell effector program contain demethylated regions in both T(CMV) and $T_{mem}$ cells, but are methylated in $T_N$ cells, including genes coding for molecules regulating T cell-mediated cytotoxicity (*GZMA*, *GZMB*, *GZMK*, *IFNG*), actin cytoskeletal organization (*ACTA2*, *ACTB*, *ACTN4*, *DOCK2*), integrin-dependent cell adhesion (*ITGB1*, *ITGB1BP1*, *ITGA5*), chemokine-mediated signalling (*CCL1*, *CCL4*, *CCL5*, *CCR4*, *CCR5*), TGFβ-mediated signalling (*SMAD2*, *SMAD3*, *SMAD7*), and WNT/β-catenin signalling (*APC2*, *CTR9*, *CYLD*, *WNT11*, *WNT7B*) (**S1 Table**). Taken

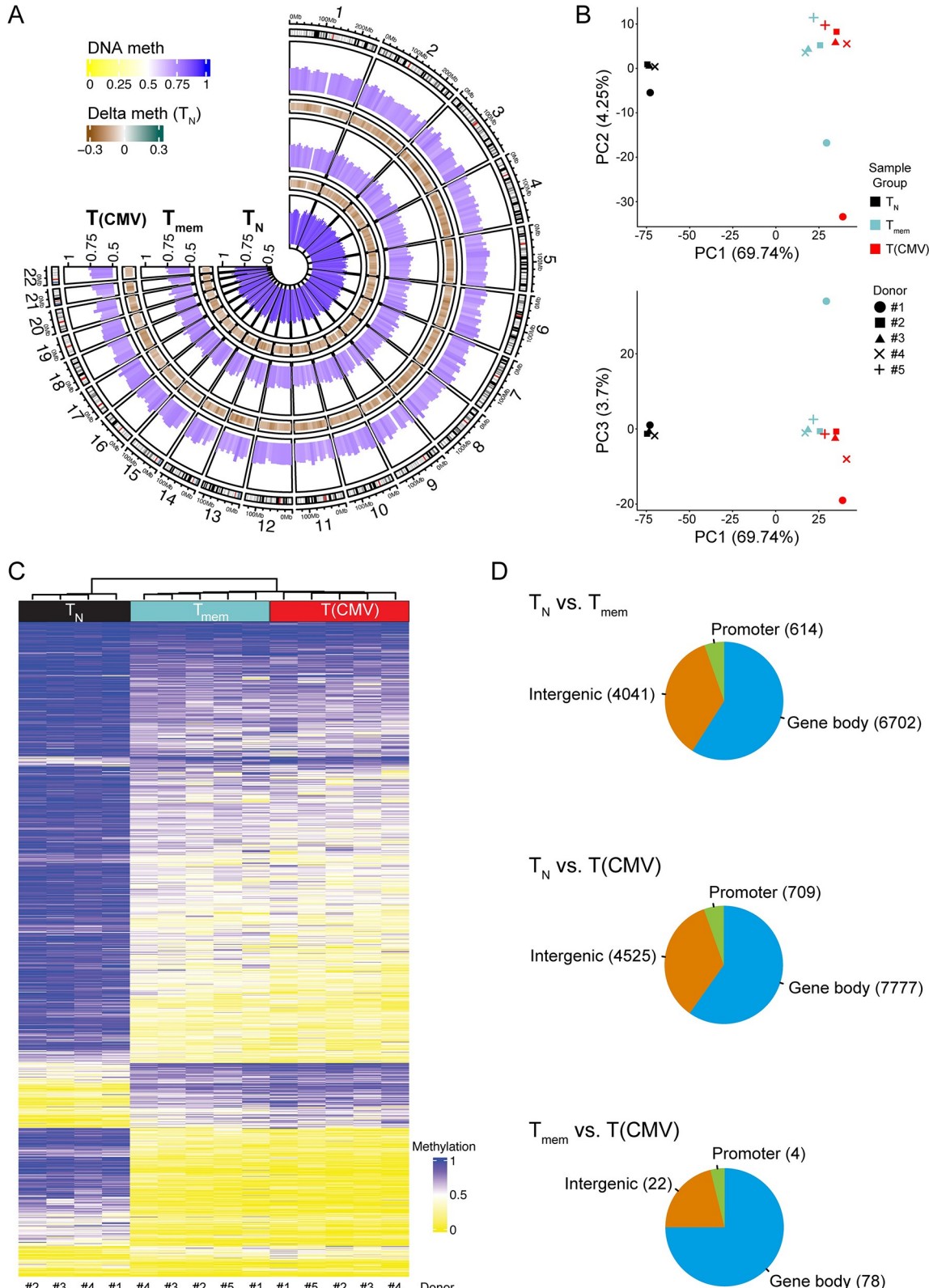

**Fig 1. Genome-wide methylation profiling of T(CMV) cells.** CD8+ T cell subsets including $T_N$ (n = 4), $T_{mem}$ (n = 5), and T(CMV) cells (n = 5) were sorted using flow cytometry. From each sample, genomic DNA was isolated and converted by bisulfite treatment, followed by WGBS. **(A)** Circos plot showing DNA methylation levels for $T_N$, $T_{mem}$, and T(CMV) cells across the whole genome (in 10

kb tiling windows, aggregated all donors). CpG methylation levels are represented as histogram tracks across the genome based on the sample. High levels of methylation are indicated by dark blue, while low levels of methylation are indicated by light yellow. Brown-teal colour-coding between the bar tracks indicates differences in methylation levels relative to $T_N$ cells. **(B)** Principal Component Analysis (PCA) per cell type and donor samples based on 50,000 highly variable 1 kb tiling regions. The percent of explained variance for each component is denoted in the axis labels. **(C)** Hierarchical clustering of 15,598 non-overlapping DMRs identified in pair-wise comparisons of methylomes from $T_N$, $T_{mem}$, and T(CMV) cells. The dendrogram on top corresponds to hierarchical clustering of the samples. The colour-code illustrates the mean methylation levels of the DMRs as indicated (yellow: methylation level = 0%, blue: methylation level = 100%). **(D)** Pie charts indicating the position of the pairwise DMRs identified in indicated group-wise comparisons relative to annotated genes. Numbers in parentheses show the number of DMRs in intergenic, promoter, or gene body regions according to their genomic position.

together, the WGBS study demonstrated substantial changes of DNA methylation patterns during the differentiation of $T_N$ into T(CMV) cells.

Next, we focused our analysis on the 104 DMRs that were identified in the comparison of T(CMV) vs. $T_{mem}$ cells. A small fraction (25 DMRs) was also detected in the comparison of $T_{mem}$ vs. $T_N$ cells, and showed a more pronounced demethylation in T(CMV) cells (**S1 Table**). However, the majority (79 DMRs) was only detected in the comparison of T(CMV) vs. $T_{mem}$ cells, and thus determine the unique epigenetic signature of T(CMV) cells. It is important to note that in a few cases more than one DMR was found in proximity to a given gene, resulting in 71 unique genes showing differential DNA methylation patterns between T(CMV) and $T_{mem}$ cells (**S1 Table**). Some of the top demethylated DMRs in T(CMV) cells, identified in the pairwise comparison with $T_{mem}$ cells, were associated to genes with known roles in differentiation and cytotoxic function of CD8+ T cells, including *KLRD1*, *TBX21*, *ZEB2*, and *S1PR5* [47–50] (**Fig 2**). In addition, a number of DMRs being selectively demethylated in T(CMV) cells were associated to genes for which a role in CD8+ T cells has not been described yet, namely *FGR*, *LINC01871*, *TMEM14C*, *MAD1L1*, *LMF1*, and *TBKBP1*. TBK-binding protein 1 (TBKBP1) is an adaptor protein that can interact with TBK1 and promote its activation [51,52]. TBK1 is a central player during the type I IFN and pro-inflammatory cytokine innate immune response by mediating phosphorylation and thereby activation of the transcription factors interferon regulatory factor (IRF) 3 and 7 and also NF-κB [53–59]. The transcription signature of CD8+ $T_{EMRA}$ cells can be regulated by several innate immune-related genes, including natural killer (NK) cell activation mediators and multiple NK receptors [33]. These findings prompted us to study the role of TBKBP1 in CD8+ T cells in more detail.

## Demethylation of the *TBKBP1* DMR is associated with higher *TBKBP1* gene expression in effector memory T cell subsets but not with alternative splicing

To characterise the methylation status of the newly identified *TBKBP1* DMR within major human CD8+ T cell subpopulations, we isolated genomic DNA from sorted CD45RA+CCR7+ $T_N$, CD45RA+CCR7− $T_{EMRA}$, CD45RA−CCR7− $T_{EM}$, CD45RA−CCR7+ $T_{CM}$, and CD45RA+CCR7+CD28+CD62L+CD95+ $T_{SCM}$ cells isolated from peripheral blood of CMV-seropositive donors and subjected them to pyrosequencing analysis. Analysis of the *TBKBP1* DMR showed a pronounced demethylation pattern in $T_{EMRA}$ cells and a partial demethylation in $T_{EM}$ cells, while $T_{CM}$, $T_{SCM}$ and particularly $T_N$ cells were largely methylated at these sites (**Fig 3A**). Since it has been previously reported that CpG methylation levels around the promoter or first intron/exon region of a specific gene inversely correlate with gene expression [60–62], we next asked if the strong demethylation of the *TBKBP1* DMR in $T_{EMRA}$ cells is accompanied by a high *TBKBP1* expression. Indeed, analysis of *TBKBP1* gene expression revealed a high expression in $T_{EMRA}$ cells, an intermediate expression in $T_{EM}$ cells, a low expression in $T_{CM}$ and $T_{SCM}$ cells, while *TBKBP1* expression was below the detection level in

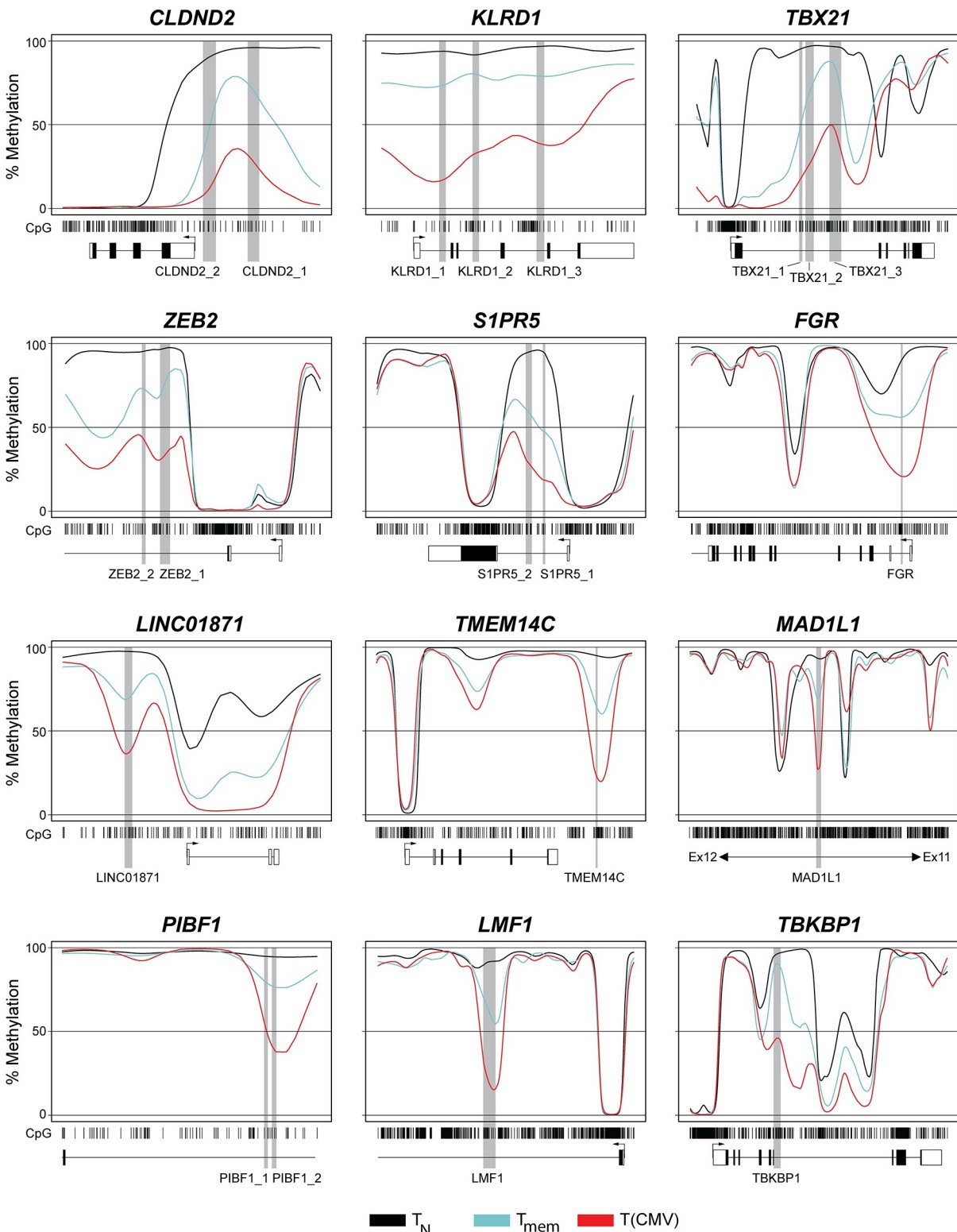

**Fig 2. Methylation profiles of gene loci associated with top DMRs from epigenetic signature of T(CMV) cells.** Out of the 71 genes showing unique differential DNA methylation patterns between T(CMV) and $T_{mem}$ cells the top 12 DMR-associated gene loci were selected. For each gene, CpG motifs (barcodes), DMRs (light grey boxes), and exons of the surrounding gene body (dark grey boxes, transcriptional start site indicated by arrows) are displayed. Coloured lines illustrate methylation values ranging from 0–100% of $T_N$ (black), $T_{mem}$ (cyan), and T(CMV) cells (red) in a linear manner. Mean values from all donors per CD8+ T cell subset are depicted.

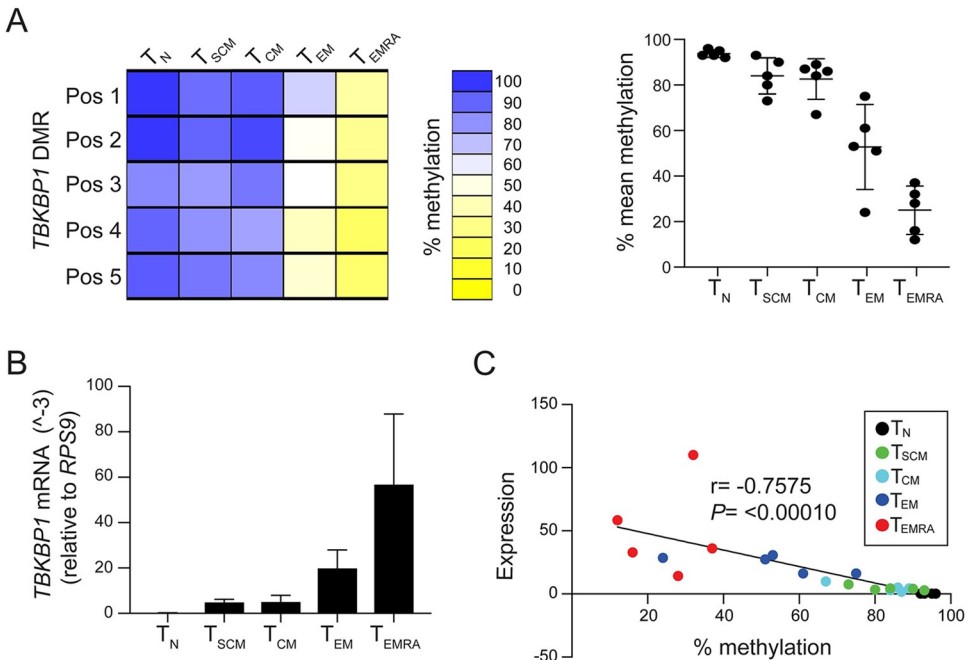

**Fig 3. Demethylation of *TBKBP1* DMR correlates with increased *TBKBP1* expression in T_EMRA and T_EM CD8⁺ T cell subsets and is stably maintained upon *in vitro* culture.** Indicated CD8⁺ T cell subsets were isolated from CMV-seropositive healthy donors and genomic DNA as well as RNA were isolated from sorted samples. Bisulfite-converted genomic DNA was subjected to pyrosequencing using primers targeting the *TBKBP1* DMR and RNA was transcribed into cDNA to determine *TBKBP1* expression levels by qRT-PCR. **(A)** Methylation profiles of the *TBKBP1* DMR in indicated CD8⁺ T cell subsets. (Left) The methylation values from 1 representative donor were translated into a colour-code according to the scale ranging from yellow (0% methylation) via white (50% methylation) to blue (100% methylation), the position of the CpG motifs is depicted, and each rectangle represents the methylation value of a single CpG motif. (Right) The scatter plot shows the mean methylation level of all 5 CpG motifs of the *TBKBP1* DMR in indicated CD8⁺ T cell subsets from 5 independent donors. Each dot represents data from one donor and mean values±SD are depicted. **(B)** The bar plot shows the relative *TBKBP1* expression normalized to the housekeeping gene *RPS9* in indicated CD8⁺ T cell subsets. Mean values±SD are depicted (n = 5). **(C)** Scatterplot and linear regression analysis show correlation of mean methylation of *TBKBP1* DMR with mean *TBKBP1* expression in indicated CD8⁺ T cell subsets from 5 donors (r, correlation coefficient; *p*, *p*-value).

T_N cells (**Fig 3B**). Hence, we could observe a clear inverse correlation between DNA methylation rates and *TBKBP1* gene expression levels (r = -0.7575; *p*-value = <0.00010) (**Fig 3C**). Interestingly, we identified a nearly identical *TBKBP1* DMR methylation (**S3A Fig**) and mRNA expression pattern (**S3B Fig**) in sorted CD8⁺ T cell subsets isolated from human CMV-seronegative donors, also leading to an inverse correlation of the two data sets (r = -0.8965, *p*-value = <0.00010) (**S3C Fig**). In conclusion, our results indicate a DMR-mediated transcriptional control of the *TBKBP1* gene locus, which seems to be independent of the CMV status.

Besides altering the accessibility of DNA-binding transcription factors and thus the rate of transcription, changes in DNA methylation can also affect splicing, leading to the generation of alternative transcripts [63]. Since the *TBKBP1* DMR is located close to an exon/intron junction (**S4A Fig**), differences in DNA methylation at the *TBKBP1* DMR might result in distinct alternative splicing of the *TBKBP1* transcripts in CD8⁺ T cell subsets. Specific RT-PCRs were designed to discriminate between the different *TBKBP1* transcripts reported in the Ensembl genome browser (**S4A Fig**). However, the statistical analysis of the *TBKBP1-A/TBKBP1-B* transcript ratios of sorted T_CM, T_EM and T_EMRA cells revealed no significant difference (**S4B Fig**), in line with a recently published study that did not detect TBKBP1 protein variants in

total CD4$^+$ and CD8$^+$ T cells as well as NKT cells [64]. Thus, the DNA methylation changes at the *TBKBP1* DMR does not result in distinct alternative splicing of the *TBKBP1* transcripts.

## Stable *TBKBP1* DMR methylation pattern during long-term *in vitro* culture of CD8$^+$ T cell subsets

During homeostatic proliferation, naive CD8$^+$ T cells have been reported to undergo phenotypic alterations [65,66]. This observation raised the question whether the methylated and demethylated states of the *TBKBP1* DMR in T$_N$ and T$_{EMRA}$ cells, respectively, remain stable during proliferation. To answer this question, we isolated CD8$^+$ T$_N$ and T$_{EMRA}$ cells from peripheral blood of CMV-seropositive healthy donors and cultured them for up to 30 days in the presence of anti-CD3 and anti-CD28 antibodies plus IL-2. Every 5 days, an aliquot was taken for phenotypic characterisation by flow cytometry and pyrosequencing. Upon culture of T$_N$ cells, we observed a transient downregulation of the homing receptor CCR7 reaching normal levels by day 20, and an acquisition of stem-like properties with the upregulation of CD62L and CD28 on day 15 (**S5 Fig**), as recently reported [41]. In contrast, T$_{EMRA}$ cells more stably maintained their CD45RA$^+$CCR7$^-$ phenotype throughout the culture. Pyrosequencing analyses revealed that the *TBKBP1* DMR remained fully methylated in cultured T$_N$ cells and fully demethylated in cultured T$_{EMRA}$ cells during the entire cultivation period (**Fig 4**). Together, these findings suggest that the epigenetic status of the *TBKBP1* gene is rather stable and preserved in both cultured T$_N$ and T$_{EMRA}$ cells even after multiple rounds of T cell receptor (TCR)-triggered cell division.

## Elevated TBKBP1 expression levels correlate with increased TBK1 phosphorylation in CD8$^+$ T cells

Although it is well known that the kinase TBK1 can be activated by various signals, including TCR signalling [53,67], the role of its adaptor protein TBKBP1 in CD8$^+$ T cells remains unidentified. To characterize the status of the TBK1 signalling pathway in human CD8$^+$ T cell subsets, we first analysed the total protein levels of TBKBP1 and TBK1 as well as phosphorylated TBK1 in T$_N$, T$_{CM}$, T$_{EM}$, and T$_{EMRA}$ cells *ex vivo* isolated from CMV-seropositive healthy donors by immunoblotting. In accordance with the *TBKBP1* mRNA expression data (**Fig 3B**), T$_N$ cells hardly showed any TBKBP1 expression, while it was clearly detectable in T$_{CM}$, T$_{EM}$, and T$_{EMRA}$ cells (**Fig 5A**). Similarly, TBK1 was only weakly expressed in T$_N$ cells (**Fig 5A**), and T$_{CM}$ cells showed slightly reduced TBK1 protein levels when compared to T$_{EM}$ and T$_{EMRA}$ cells, albeit not reaching statistical significance (**Fig 5B**). However, T$_{EMRA}$ cells contained significantly more phosphorylated TBK1 when compared to T$_{CM}$ cells (**Fig 5C**), indicating a higher basal activity of the TBK1 signalling pathway in terminally differentiated CD8$^+$ T cells.

In a second step, we sought to get further insights into the role of TBKBP1 for the phosphorylation of TBK1. To this end, we retrovirally overexpressed TBKBP1 in CD8$^+$ T cells and subsequently monitored TBK1 phosphorylation after short-term restimulation with anti-CD3 and anti-CD28 antibodies (**S6A Fig**). Successfully transduced CD8$^+$ T cells were sorted by flow cytometry (**S6B Fig**) and TBKBP1-transdcued cells showed high *TBKBP1* mRNA expression levels when compared to EV-transduced controls as expected (**S6C Fig**). In addition, immunoblotting analysis confirmed high TBKBP1 protein expression in CD8$^+$ T cells retrovirally transduced with the TBKBP1 overexpression construct, and no differences in total TBKBP1 expression were observed between unstimulated and short-term restimulated CD8$^+$ T cells (**Fig 6A**). Similarly, neither TBKBP1 overexpression nor the short-term restimulation influenced total TBK1 expression in CD8$^+$ T cells (**Fig 6B**). Notably, while EV-transduced control cells did not show any increase in TBK1 phosphorylation when short-term restimulated cells

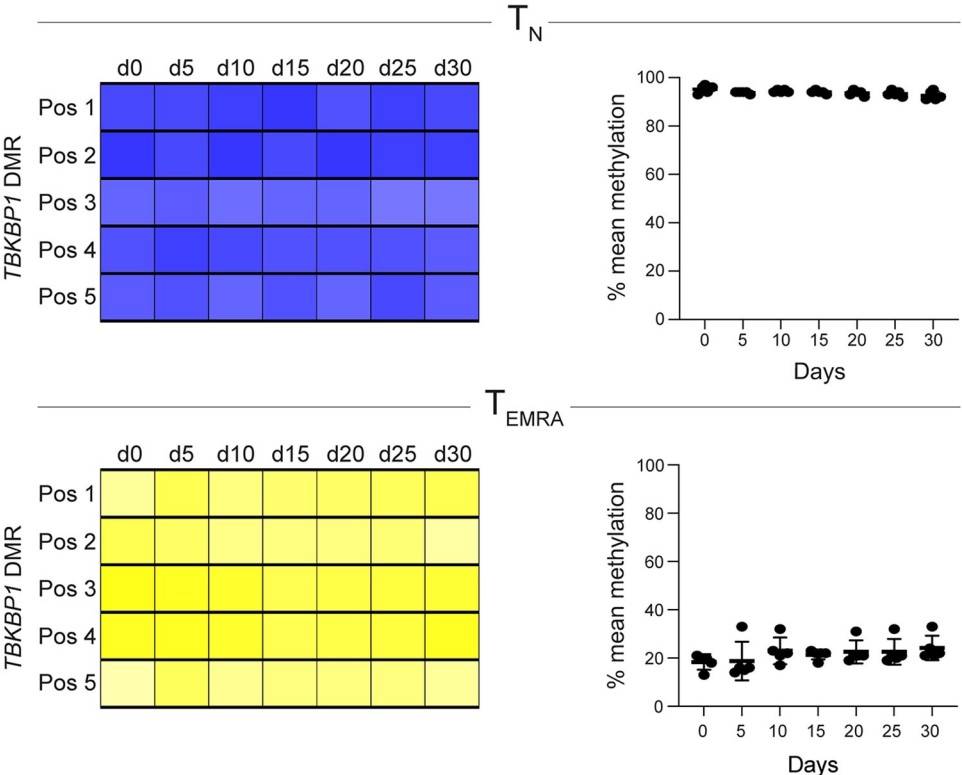

**Fig 4. Long-term *in vitro* culture of CD8+ T cell subsets does not alter the *TBKBP1* DMR methylation patterns.**
CD8+ T_N and T_EMRA cells were isolated from CMV-seropositive healthy donors and cultured *in vitro* with plate-bound anti-CD3/CD28 antibodies in the presence of exogenous human IL-2 for up to 30 days. Every 5 days, samples from cultured T_N (top) and T_EMRA cells (bottom) were taken to analyse the *TBKBP1* DMR methylation status as described in Fig 3. The heatmaps are from a representative donor (left) and scatter plots summarize the mean methylation levels of all 5 CpG motifs of the *TBKBP1* DMR from 5 independent donors. Each dot represents data from one donor and mean values±SD are depicted.

were compared to unstimulated controls, the phosphorylation of TBK1 was significantly enhanced in TBKBP1-overexpressing CD8+ T cells after short-term restimulation (**Fig 6C and 6D**). Since a recent report had demonstrated that TBKBP1 can support the recruitment of TBK1 to protein kinase C-theta (PKCθ) in murine lung epithelial cells, thereby promoting TBK1 phosphorylation and activation [52], we here also investigated the role of PKCθ for the TBK1 signalling pathway in CD8+ T cells. First, we assessed the impact of PKCθ on the phosphorylation of TBK1 by adding a highly selective PKCθ inhibitor (HY-112681) to EV-transduced or TBKBP1-overexpressing CD8+ T cells during short-term restimulation. As already mentioned above, immunoblotting analysis confirmed high TBKBP1 protein expression in TBKBP1-overexpressing CD8+ T cells, and the TBKBP1 overexpression did not influence total TBK1 expression (**S7 Fig**). Notably, upon addition of the PKCθ inhibitor we did not observe any effect on the phosphorylation of TBK1, neither in TBKBP1-overexpressing nor in EV-transduced control cells (**S7 Fig**), suggesting that PKCθ is not promoting the phosphorylation of TBK1 in CD8+ T cells under the tested stimulation conditions. However, the PKCθ inhibitor had an impact on the functional properties of TBKBP1-overexpressing CD8+ T cells and caused reduced IFN-γ expression levels after a 4-hour restimulation period when compared to DMSO-treated controls (**Fig 6E**). Interestingly, this inhibitory effect was only observed in TBKBP1-overexpressing CD8+ T cells and not in EV-transduced controls. Taken together, our

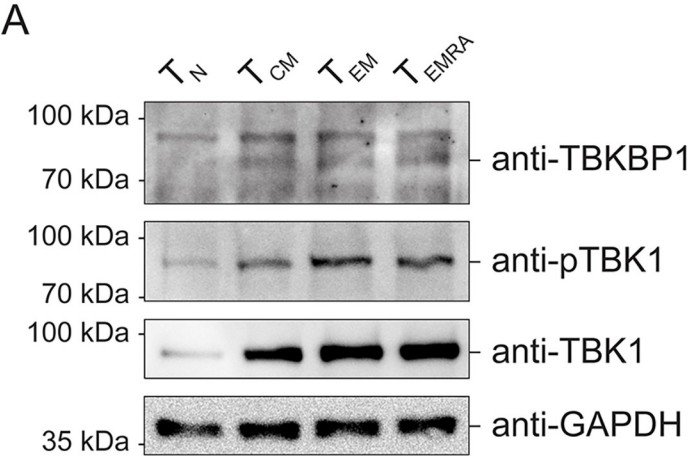

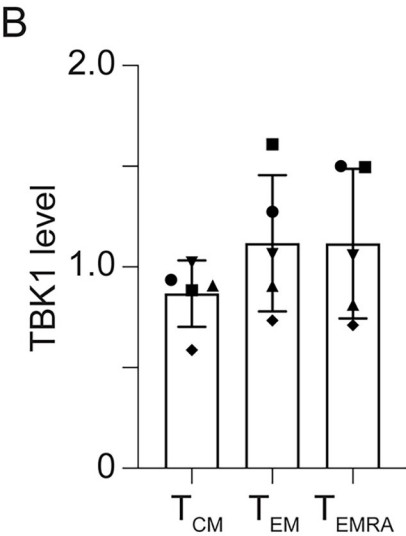

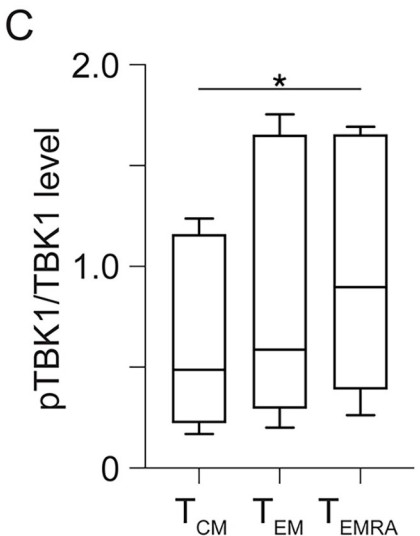

**Fig 5. $T_{EM}$ and $T_{EMRA}$ cells display an increased phosphorylation of TBK1 when compared to $T_{CM}$ cells.** To characterize the status of the TBK1 pathway in CD8⁺ T cell subsets, $T_N$, $T_{CM}$, $T_{EM}$ and $T_{EMRA}$ cells were isolated from CMV-seropositive healthy donors, and lysates were generated and utilised for immunoblotting. **(A)** Representative immunoblot analysis of indicated CD8⁺ T cell subsets from one out of five donors showing the expression of TBKBP1, phosphorylated TBK1 (pTBK1) and TBK1. The analysis of GAPDH expression served as loading control. **(B)** The bar plots show GAPDH- or α-tubulin-normalized band intensities for TBK1 quantified from sorted $T_{CM}$, $T_{EM}$ and $T_{EMRA}$ cells of 5 donors. Symbols indicate samples from the same donor. **(C)** The box-and-whiskers plots show the ratio of GAPDH- or α-tubulin-normalized band intensities of phosphorylated TBK1 and TBK1 from sorted $T_{CM}$, $T_{EM}$ and $T_{EMRA}$ cells of 5 donors. For statistical analyses, a paired two-tailed student's t test was conducted with *, $p \leq 0.05$.

findings suggest that TBKBP1 expression can promote TBK1 phosphorylation in activated CD8⁺ T cells in a PKCθ-independent manner, likely resulting in the activation of further downstream signalling pathways.

## Ectopic expression of TBKBP1 augments the virus-reducing capacity of CD8⁺ T cells

After having demonstrated that TBKBP1 overexpression has an effect on TBK1 phosphorylation, we next sought to investigate its functional role in CD8⁺ T cells during an immune reaction. To this end, we utilized ARMATA, an assay for rapid measurement of antiviral T cell

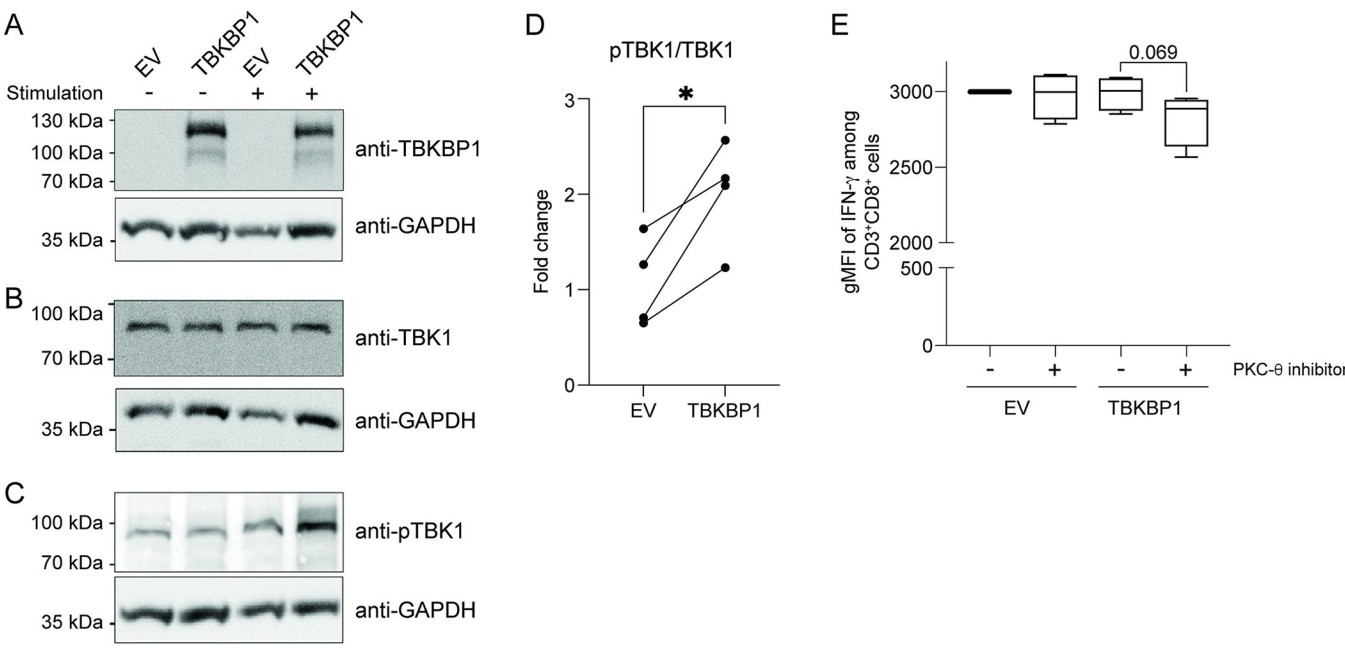

**Fig 6. TBKBP1 overexpression in restimulated CD8⁺ T cells results in increased TBK1 phosphorylation.** PBMCs obtained from CMV-seronegative healthy donors were stimulated with plate-bound anti-human CD3 and anti-human CD28 for 48 hours and subsequently retrovirally transduced with TBKBP1-expressing vectors or empty vector (EV) controls. Successfully transduced CD8⁺mCherry⁺ T cells were sorted by flow cytometry, serum-starved overnight, and subsequently stimulated with soluble anti-human CD3, anti-human CD28, and cross-linking antibodies, while unstimulated cells served as additional controls. Samples from both unstimulated (-) and stimulated (+) EV-transduced or TBKBP1-overexpressing CD8⁺ T cells were subjected to immunoblotting to determine the expression of **(A)** TBKBP1, **(B)** TBK1, and **(C)** phosphorylated TBK1 (pTBK1). The analysis of GAPDH expression served as loading control. **(D)** Band intensities for pTBK1 and TBK1 were quantified and the fold change of TBK1 phosphorylation upon stimulation of EV-transduced or TBKBP1-overexpressing CD8⁺ T cells was determined by dividing the ratio of pTBK1/TBK1 band intensities of stimulated cells to the ratio of pTBK1/TBK1 band intensities of unstimulated samples. Data are combined from 4 independent donors analysed in 2 independent experiments. **(E)** EV-transduced or TBKBP1-overexpressing CD8⁺ T cells were stimulated in the presence of a PKCθ inhibitor (+) or treated with DMSO as control (-). Subsequently, CD3⁺CD8⁺ cells were analysed for intracellular IFN-γ expression by flow cytometry. The geometric mean fluorescent intensity (gMFI) of IFN-γ expression among CD3⁺CD8⁺ cells is depicted. Data are combined from 4 independent donors analysed in 2 independent experiments. **(D, E)** For statistical analyses, a paired two-tailed student's t test was conducted with *, $p \leq 0.05$.

activity [68]. In this assay, fibroblasts (MRC-5 cells) were infected with recombinant CMV expressing the fluorescent reporter mNeonGreen [69] and co-cultured with CMV-specific CD8⁺ T cells, which had been generated by retroviral overexpression of the human leukocyte antigen (HLA)-A*02:01-restricted high-avidity TCR recognising the CMVpp65-derived peptide NLVPMVATV (mTCR 5–2) [70]. Additionally, to assess the functional role of TBKBP1, cells were retrovirally transduced with the TBKBP1 overexpression construct, while EV-transduced cells served as control. Successfully transduced mTCR 5–2⁺mCherry⁺ CD8⁺ T cells were sorted by flow cytometry (**S8 Fig**) and added to the CMV-infected MRC-5 cells at different effector:target (E:T) ratios. The ARMATA measures the viral expression of mNeonGreen by temporal live cell imaging in 1-hour intervals and capture of representative microscopic images. As shown in **Fig 7A**, we observed a strong antiviral and cytopathic effect of mTCR 5–2⁺ CD8⁺ T cells, with an overall reduced number of viable mNeonGreen-expressing MRC-5 cells when compared to controls lacking T cells. Hourly measurements of the reporter signal revealed a continuous increase of the reporter signal in cultures without T cells, reaching a plateau around 12 hours with an only mild increase afterwards (**Fig 7B**, green curve). Upon addition of EV-transduced mTCR 5–2⁺ control CD8⁺ T cells, a marked decrease of the reporter signal was observed, starting at 12 hours and continuing until the end of the assay at 72 hours (**Fig 7B**, blue curve). This signal reduction was even more pronounced when TBKBP1-overexpressing mTCR 5–2⁺ CD8⁺ T cells were added, showing a significantly stronger neutralization of MRC-5-infected cells irrespective of the E:T ratio (**Fig 7B**, red curve). The improved virus reduction by TBKBP1-overexpressing CD8⁺ T cells was associated with significantly elevated levels of IFN-γ and Granzyme A, and a trend towards increased levels of Granzyme B, Perforin, MIP-1α (CCL3), and MIP-1β (CCL4) in supernatants of co-cultures with TBKBP1-overexpressing CD8⁺ T cells when compared to EV-transduced controls (**Figs 7C and S9**). Collectively, these data demonstrate that ectopic expression of TBKBP1 in human CD8⁺ T cells results in the upregulation of pro-inflammatory cytokines and chemokines, facilitating a significantly enhanced virus neutralization capacity.

## Discussion

Epigenetic processes, including DNA methylation, are well-known contributors to T cell fate specification upon CD8⁺ T cell differentiation [23–25]. In the present study, we hypothesized that global DNA methylation analyses could provide insights into the epigenetic reprogramming of T(CMV) cells and also support the identification of key players involved in their effector functions. Thus, we compared the DNA methylation profile of CMVpp65-reactive T(CMV) cells with the DNA methylation profile of antigen-experienced CD8⁺ T cells (CCR7⁻CD28^{high}CD27⁺CD45RA⁻) with unknown specificity. Utilizing WGBS, we observed substantial epigenetic alterations upon T cell differentiation, and the vast majority of identified DMRs were demethylated in both $T_{mem}$ and T(CMV) when compared to $T_N$ cells. With the final aim to identify an epigenetic signature of T(CMV) cells, we focused our analysis on the comparison of DNA methylation profiles of $T_{mem}$ and T(CMV) cells, and could identify 79 regions being differentially methylated between T(CMV) and $T_{mem}$ cells and not between $T_{mem}$ and $T_N$ cells, thus representing a unique epigenetic signature of T(CMV) cells. One of these epigenetic signature genes was *TBKBP1*, not described in the context of T cell-mediated immune responses so far. The identified *TBKBP1* DMR was found upstream of exon 6, and it is tempting to speculate that transcription factors binding to this DMR might co-operate with the known *TBKBP1* enhancer region located between exon 4 and 5 (Ensembl ENSR00001010382).

The pronounced demethylation of the *TBKBP1* DMR in T(CMV) cells encouraged us to investigate the specific role of TBKBP1 for the cytotoxic function of CD8⁺ T cells. We could

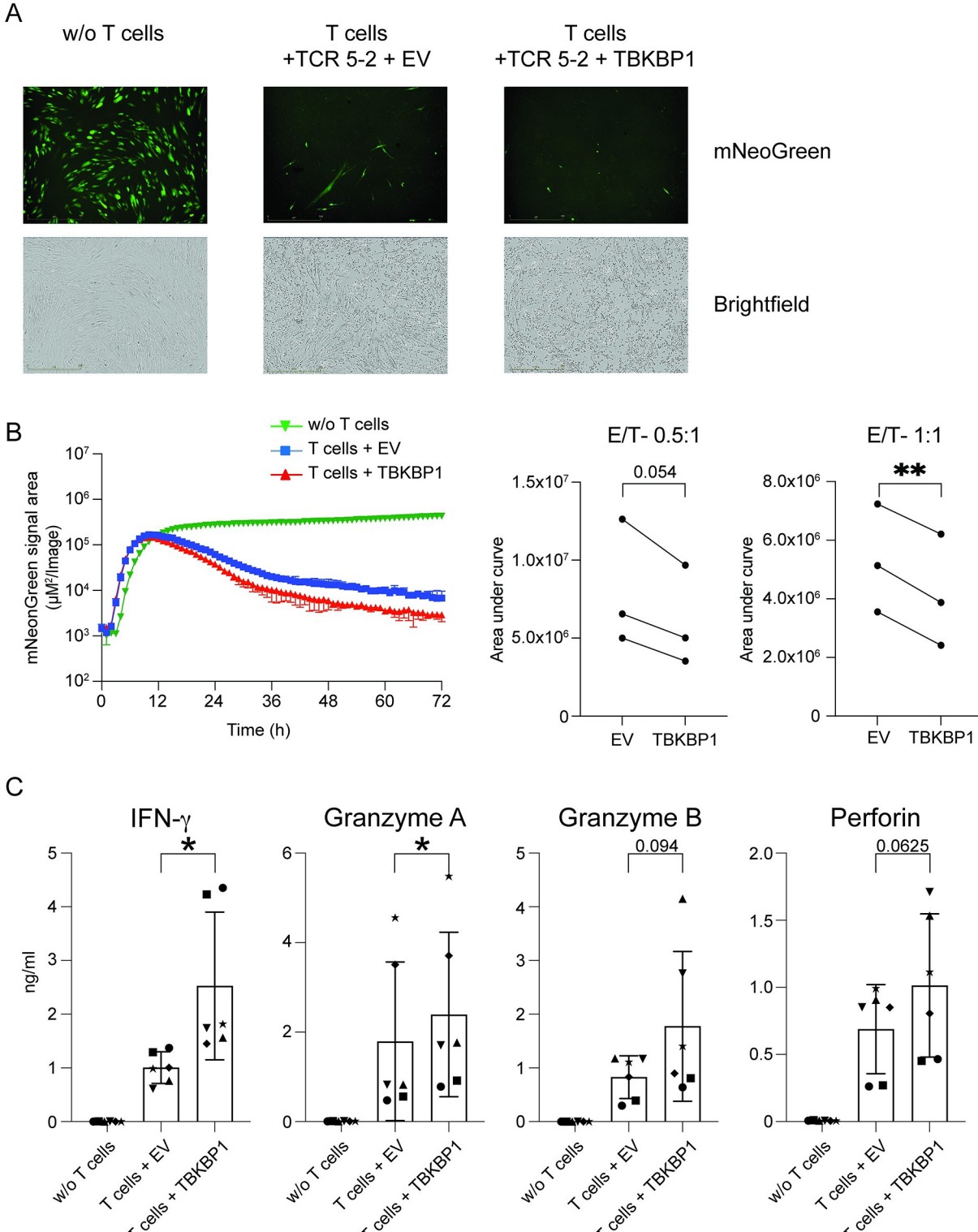

**Fig 7. TBKBP1 overexpression enhances virus-reducing capacity of CD8+ T cells.** PBMCs obtained from CMV-seronegative healthy donors were stimulated with plate-bound anti-human CD3 and anti-human CD28 for 48 hours and subsequently retrovirally transduced with both the high-avidity TCR 5–2 containing murine constant regions (mTCR) and directed against the CMV HLA-A*02-peptide NLVPMVATV and also TBKBP1-overexpressing vectors or empty vector (EV) controls. Successfully transduced mTCR 5–2+mCherry+ CD8+ T cells were sorted by flow cytometry and added at indicated effector:target (E:T) ratios to MRC 5 cells infected with recombinant CMV expressing the reporter

protein mNeonGreen. ARMATA measures the expression of mNeonGreen using live cell imaging. **(A)** Representative microscopic images 48 hours after infection showing the expression of mNeonGreen (fluorescence image) and the location of the cells in corresponding brightfield images in CMV-infected MRC-5 cells in the absence of added T cells (w/o T cells), in the presence of EV-transduced mTCR 5–2+ CD8+ T cells (T cells + TCR 5–2 +EV) or in the presence of TBKBP1-overexpressing mTCR 5–2+ CD8+ T cells (T cells + TCR 5–2 +TBKBP1). T cells were added at a E:T ratio of 1:1. **(B)** The ARMATA was quantified by hourly measurements of the reporter signal in cultures in the absence of added T cells (w/o T cells, green curve), in the presence of EV-transduced mTCR 5–2+ CD8+ T cells (T cells + TCR 5–2 +EV, blue curve) or in the presence of TBKBP1-overexpressing mTCR 5–2+ CD8+ T cells (T cells + TCR 5–2 +TBKBP1, red curve). (Left) The graph depicts mean values ±SD from 2 technical replicates of 1 representative out of 3 independent experiments. (Right) The line plots summarize data from 3 independent experiments by showing the average area under the curve (AUC) ±95% CI for EV-transduced or TBKBP1-overexpressing mTCR 5–2+ CD8+ T cells cultured with CMV-infected MRC-5 at indicated E:T ratios. **(C)** 36 hours after infection, culture supernatants were harvested and cytokine profiles were determined from cultures of CMV-infected MRC-5 cells in the absence of added T cells (w/o T cells), in the presence of EV-transduced mTCR 5–2+ CD8+ T cells (T cells + EV, E:T = 1:1) or in the presence of TBKBP1-overexpressing mTCR 5–2+ CD8+ T cells (T cells + TBKBP1, E:T = 1:1). Data from 3 independent experiments with 2 technical replicates each are shown. Symbols indicate samples from the same donor. For statistical analyses, a paired two-tailed student's $t$ test (leaving out "w/o T cell" group) was conducted with *, $p \leq 0.05$ and **, $p \leq 0.01$.

validate the preferential demethylation of the *TBKBP1* DMR in $T_{EMRA}$ cells, the dominant phenotype of T(CMV) cells, which also showed the highest *TBKBP1* mRNA expression levels and the highest levels of phosphorylated TBK1, suggesting a higher basal activity of the TBK1 signalling pathway in terminally differentiated CD8+ T cells. Yet, the demethylation of the *TBKBP1* DMR, and probably the increased activity of the TBK1 signalling pathway, does not seem to be restricted to CMV-specific T cells, since we also found a pronounced *TBKBP1* DMR demethylation correlating with enhanced *TBKBP1* mRNA expression in $T_{EMRA}$ cells from CMV-seronegative donors. Since other viruses, such as dengue, are known to promote the formation of $T_{EMRA}$ cells [71], our findings suggest that the observed epigenetic regulation of *TBKBP1* expression could be also caused by immune responses to other pathogens.

A direct involvement of TBKBP1 in the phosphorylation of TBK1 was confirmed using TBKBP1-overexpressing cells, which exhibited a significantly increased TBK1 phosphorylation upon TCR triggering. This suggests that TBKBP1 is crucial for the optimal TCR-induced activation of TBK1 and downstream signalling pathways. Accordingly, TBKBP1 overexpression resulted in an increased production of pro-inflammatory cytokines and chemokines and promoted the cytotoxicity of T(CMV) cells against CMV-infected target cells. Collectively, these observations demonstrate the crucial role of TBKBP1 for optimal adaptive T cell responses and contribute to our current understanding of CD8+ T cell responses to CMV. However, it must be noted that overexpression systems might cause aberrant effects when compared to endogenous expression of a given protein. Thus, to further understand the role of TBKBP1 for the TBK1 signalling pathway and antiviral mechanisms in CD8+ T cells, the impact of endogenous TBKBP1 expression requires further investigation.

The role of DNA methylation for T cell fate specification has mostly been studied in relation to cytokines and effector molecules [41,72,73]. More recently, a number of genome-wide epigenome profiling studies gave first insights into the global changes upon T cell differentiation [23,25,27,28,35,36,45]. A seminal comparative study on acute and chronic viral infections demonstrated that naive antigen-specific CD8+ T cells are epigenetically regulated and differentiate rapidly into either a memory or exhaustive phenotype upon infection [73]. Interestingly, a recent report showed that even during their early phase of differentiation, CD8+ T cells responding to acute and chronic infections displayed distinct transcriptional and epigenetic landscapes [74]. Among the unique epigenetic signature genes of T(CMV) cells identified in the present study, *KLRD1*, *TBX21* (coding for T-bet), and *ZEB2* have already been reported to regulate functional properties of memory CD8+ T cells and were found to be upregulated in chronic infections or exhausted CD8+ T cells [14,49,74–78]. Another gene locus, *ADAM28*, which was recently identified in an ATACseq-based epigenetic profiling study comparing human $T_{EMRA}$ with $T_{CM}/T_{EM}$ cells [79], was not among the unique epigenetic signature genes

of T(CMV) cells, but only found when T(CMV) were compared to $T_N$ cells. These findings suggest that small differences in sorting strategies to isolate CD8⁺ T cell subsets or the use of different epigenetic profiling methods might result in different unique epigenetic signature genes.

Furthermore, we noted that DMRs linked with *S1PR5*, a migratory receptor that is expressed by circulating memory CD8⁺ T cells, but selectively downregulated in tissue-resident memory CD8⁺ T cells [50,80,81], were demethylated in T(CMV) cells. A recent report on murine tissue-resident memory T cells demonstrated that *S1pr5* induction is directly controlled by T-bet and Zeb2 [50]. Additionally, CD8⁺ T cells from chronic infection models of lymphocytic choriomeningitis virus exhibit a higher frequency of T-bet⁺ cells and higher T-bet expression levels than those from acute infection models [74]. These reports further strengthen the findings from our DNA methylation dataset and suggest that in T(CMV) cells, the selectively demethylated genes *TBX21*, *ZEB2*, and *S1PR5* may jointly coordinate the effector function of antigen-specific CD8⁺ T cells.

Moreover, we identified a DMR associated with *TBKBP1* to be strongly demethylated in T(CMV) cells. TBKBP1 was initially reported to be engaged in the TNFα/NF-κB pathway to elicit innate immune responses [51]. The antiviral response of the innate immune system relies on various inducible transcription factors such as NF-κB and IRFs, which account for the production of IFNs and pro-inflammatory cytokines [82–84]. Studies showed that IκB kinase-I (IKKε), TNFR-associated factor (TRAF)-dependent NF-κB activator, and TBK1 trigger the phosphorylation of IRF3 and IRF7, which are responsible for transactivating type I and type II IFNs [54,55]. A recent report demonstrated that TBK1 can be recruited by TBKBP1 to interact with protein kinase C-theta (PKCθ), thereby enabling TBK1 phosphorylation and activation [52]. Yet, using a highly selective PKCθ inhibitor, we could not observe any impact of PKCθ on the phosphorylation of TBK1. However, the findings from the overexpression approach support a role of TBKBP1 in CD8⁺ T cells, which upon restimulation showed an enhanced TBK1 phosphorylation that could potentially support downstream signalling pathways. Moreover, various reports showed that phosphorylation of multiple serine residues within regulatory regions of IRF3 and IRF7 orchestrates TBK1 and IκB-mediated signalling pathways [85,86], that IRF7 can promote the expression of IFN-γ-responsive genes through the initiation and stabilization of RNA polymerase II recruitment in the presence of TBK1 and/or IKKε [85], and that IRF3 affects granzyme B expression and maintenance of memory T cell function in response to viral infection [87]. These findings are in line with data from the present study demonstrating that ectopic expression of TBKBP1 enhances the cytotoxic activity of CD8⁺ T cells through the upregulation of several pro-inflammatory mediators such as IFN-γ, granzyme A, granzyme B, and perforin. Collectively, our data indicate that TBKBP1 promotes the activation of TBK1 and further downstream pathways for optimal effector function of CMV-specific CD8⁺ T cells.

In conclusion, our findings provide evidence for the potential utility of TBKBP1 as a therapeutic target for the modulation of CD8⁺ T cell-mediated immune responses. Clinical relevance of our findings lies in the fact that ectopic expression of TBKBP1 enhanced the cytotoxicity of T(CMV) cells, suggesting that TBKBP1 could be exploited as a potential therapeutic target for the improvement of adoptive T cell therapy in CMV-infected individuals.

## Material and methods

### Ethics statement

All human samples were obtained from the Institute of Transfusion Medicine and Transplant Engineering, Hannover Medical School (MHH). Written informed consent was obtained

from all donors (Ethics Committee MHH, approval numbers: 3639–2017, 9001_BO-K, 9255_BO_K_2020).

## Human donors

All human samples were obtained from male healthy donors. For the phenotypic characterisation and isolation of T(CMV) cells, CMV-seropositive HLA-typed donors showing >0.5% CMV phosphoprotein 65 (pp65)-specific T cells upon antigen-specific restimulation [88,89] were selected. The PKCθ inhibition experiments and the ARMATA were performed using cells from male CMV-seronegative, HLA-A*02 negative donors. *In vitro* cultivation experiments were conducted using cells from male CMV-seropositive donors regardless of their HLA type. All remaining experiments were performed with samples from both CMV-seronegative and -seropositive donors regardless of their HLA type. CMV-serological testing and HLA-typing were conducted at the Institute of Transfusion Medicine and Transplant Engineering, MHH.

## Human sampling and cell isolation

For the phenotypic characterisation and whole genome bisulfite sequencing (WGBS) of T (CMV) cells, PBMCs were isolated from residual blood samples from disposable kits used during routine platelet apheresis at the Institute of Transfusion Medicine and Transplant Engineering, MHH, via discontinuous gradient centrifugation and immediately processed. For all remaining experiments, PBMCs were isolated from Leukocyte Reduction System (LRS) cones using Ficoll-based density gradient centrifugation (Lymphoprep; STEMCELL Technologies) and SepMate tubes (STEMCELL technologies). PBMCs were cryopreserved in FCS containing 10% DMSO. One day prior to the experiment, cryopreserved PBMCs were thawed, rinsed with excess TexMACS medium (Miltenyi Biotec) to remove DMSO, and rested overnight with TexMACS medium supplemented with 100 IU/ml of recombinant human IL-2 (Miltenyi Biotec).

## Antibodies for flow cytometry

All monoclonal antibodies used for immunophenotyping and cell sorting were purchased from BioLegend and BD Biosciences. Characterisation of T(CMV) cells was performed using anti-CD3 BV510, anti-CD8 FITC, anti-IFN-γ PE, anti-CD45RA PerCP/Cy5.5, and anti-CD62L AF647. For WGBS, T(CMV) cells and control T cell subsets were isolated using anti-CD3 AF488, anti-CD4 APC-Cy7, anti-CD8 APC, anti-CD27 BV510, anti-CD28 PE-Dazzle, anti-CD45RA PerCP/Cy5.5, anti-KLRG1 PE/Cy7, anti-CCR7 BV421, and anti-CX3CR1 BV650. For pyrosequencing, quantitative real-time-PCR (qRT-PCR), and *in vitro* cultivation experiments, CD8+ T cell subsets were isolated using anti-CD3 AF488, anti-CD8 APC, anti-CD28 PE Texas Red, anti-CD45RA PerCP/Cy5.5, anti-CD62L PE-Cy7, anti-CD95 PE, and anti-CCR7 BV421. For the isolation of retrovirally-transduced CMV-specific CD8+ T cells, anti-CD8 APC and anti-mouse TCR APC-Cy7 along with LIVE/DEAD Fixable Blue Dead Cell dye were used (ThermoFischer Scientific).

## Flow cytometry

PBMCs were prepared for flow cytometry assay as described previously [90]. Briefly, for surface staining, cells were resuspended in staining buffer (PBS and 0.5% BSA), and single-cell suspensions were labelled with antibodies for 30 minutes at 4˚C, while chemokine receptor staining was carried out at 37˚C for 30 minutes. For intracellular staining of IFN-γ, the Foxp3

staining kit (eBioscience) was used according to the manufacturer's protocol. RatIgG (Dianova, final concentration of 40 μg/ml) was added to block unspecific binding. Following staining, cells were washed and resuspended in staining buffer. Samples were acquired on a FACSCanto10c, LSR-II flow cytometer (BD Biosciences) or LSRFortessa flow cytometer (BD Biosciences). Data were analysed using BD FACSDiva software version 8.0.1 and FlowJo software version 10 (both from BD Biosciences). Cell sorting of CD8+ T cell subsets and transduced cells was performed on a BD FACS ARIA II SORP (BD Biosciences).

## Phenotypic characterisation of T(CMV) cells

For the phenotypic characterisation of T(CMV) cells, CD8+ T cells were enriched from PBMCs using the untouched CD8+ T Cell Isolation Kit (Miltenyi Biotec) according to the manufacturer's instructions. Next, the enriched CD8+ T cells were stimulated with the CMVpp65 overlapping peptide pool PepTivator CMVpp65 (Miltenyi Biotec) at a concentration of 1 μg/ml of each peptide for 4 hours. Subsequently, IFN-γ-secreting cells were detected using IFN-γ Secretion Assay/Detection Kit (PE) (Miltenyi Biotec) according to the manufacturer's instructions and subsequently phenotypically characterised by flow cytometry.

## Isolation of CD8+ T cell subsets for WGBS, pyrosequencing, qRT-PCR, and *in vitro* cultivation experiments

For the isolation of CD8+ T cell subsets for WGBS, PBMCs were first enriched for CD8+ T cells and subsequently stimulated with CMVpp65 overlapping peptide pool as described above. After the detection of IFN-γ-secreting cells, cells were stained and T(CMV) cells were isolated by flow cytometry-based cell sorting as $CD3^+CD4^-CD8^+IFN-\gamma^+$ cells, while memory CD8+ T cells ($T_{mem}$; $CD3^+CD4^-CD8^+CCR7^-CD28^{high}CD27^+CD45RA^-$) and naive CD8+ T cells ($T_N$; $CD3^+CD4^-CD8^+CCR7^+CD28^{int}KLRG1^-CX3CR1^-CD45RA^{high}$) were sorted as controls.

For pyrosequencing, qRT-PCR and *in vitro* cultivation experiments, human CD8+ T cell subsets were isolated from cryopreserved PBMCs. First, CD8+ T cells were enriched using human anti-CD8 MicroBeads and the automated magnetic activated cell sorting (autoMACS) system (both Miltenyi Biotec) according to the manufacturer's instructions. Subsequently, the CD8+ T cells were stained and sorted into $T_N$ ($CD45RA^+CCR7^+$), $T_{SCM}$ ($CD45RA^+CCR7^+CD28^+CD62L^+CD95^+$), $T_{CM}$ ($CD45RA^-CCR7^+$), $T_{EM}$ ($CD45RA^-CCR7^-$), and $T_{EMRA}$ ($CD45RA^+CCR7^-$) $CD8^+CD3^+$ T cell subsets by flow cytometry.

## Whole-genome bisulfite sequencing (WGBS)

For WGBS, sorted $T_N$, $T_{mem}$, and T(CMV) cells from the blood of various donors were used to prepare genomic DNA with the DNeasy Blood & Tissue Kit (Qiagen). Approximately 50 ng genomic double-stranded DNA per donor was converted with sodium bisulfite using the EZ DNA Methylation-Lightning Kit (Zymo Research) and fragmented by sonication (Covaris S220, 10% duty cycle, 175W peak incident power, intensity 5, 200 cycles per burst, 120 seconds). The fragmented DNA served as input for the Accel-NGS Methyl-Seq DNA Library Kit (Swift Biosciences) and resulted in libraries that were sequenced on an Illumina NovaSeq 6000 with depths between 216 and 298 million paired end reads (2 x 100 bp).

Sequencing data were processed using the nf-core/methylseq pipeline (version 2.2.0), using default parameters, genome assembly GRCh38 and—accel = true [91] (software freely accessible at http://doi.org/10.5281/zenodo.1343417). Briefly, the pipeline employs FASTQC (version 0.11.9), trimgalore (version 0.6.7) and bismark (version 0.24.0) [92] for read-level quality

control, adapter trimming, bisulfite-aware alignment, and cytosine-level DNA methylation quantification.

Methylation calls were further processed using RnBeads (version 2.17.0) [93] using a minimum per CpG coverage of 2 and removing CpGs that were covered in less than 50% of the samples, that overlapped annotated SNPs, or that were located on sex chromosomes. Genome-wide methylation values were computed by coverage weighted aggregation of CpG-level methylation values across samples in groups for 1 kb and 10 kb tiling windows. Dimensionality reduction using principal component analysis (PCA) was performed on those 50,000 1 kb tiling windows that exhibited the highest variance in aggregate methylation levels across the dataset.

Pairwise DMRs between the groups were identified using Bsmooth/BSseq (version 1.30.0) [94]. Briefly, we applied the Bsmooth() command with default parameters on unfiltered CpG methylation calls. Subsequently, only CpGs with a coverage of at least 5 in two or more samples per group were retained. Based on these filtered CpGs, we computed *t*-statistics and DMRs using BSmooth.tstat (. . ., estimate.var = "same", local.correct = TRUE) and dmrFinder (. . ., qcutoff = c(0.01,0.99), maxGap = 200). Gene relations to nearest genes were annotated using GENCODE (version 44). Finally, DMRs containing at least 3 CpGs were selected using an absolute mean methylation level difference of 25%.

Sequencing data and methylation levels reported in this paper were uploaded to GEO under accession number GSE245832. Identified DMRs are listed in **S1 Table**.

### *In vitro* cultivation of CD8+ T cell subsets

For *in vitro* cultivation, sorted $T_N$ and $T_{EMRA}$ cells were stimulated with plate-bound anti-human CD3 (OKT3; 1 μg/ml; BioLegend) and anti-human CD28 (CD28.2; 0.5 μg/ml; BioLegend) antibodies and cultured in TexMACS medium supplemented with 100 IU/ml human IL-2 at 37°C with 5% $CO_2$. Every 5 days during the 30-day culture period, cells were harvested and washed, and an aliquot was taken for pyrosequencing and phenotypic characterisation by flow cytometry. The remaining cells were re-stimulated with plate-bound anti-human CD3 and anti-human CD28 (both 0.5 μg/ml) antibodies and cultured in fresh TexMACS medium supplemented with 100 IU/ml human IL-2.

### Pyrosequencing of the *TBKBP1* DMR

Genomic DNA from *ex vivo* isolated CD8+ T cell subsets or *in vitro* cultured cells was extracted using the DNeasy Blood & Tissue Kits (Qiagen) and subsequently bisulfite-converted by using the EZ DNA Methylation-Lightning Kit (Zymo Research) according to the manufacturers' recommendations. *TBKBP1* DMR was pyrosequenced as described previously [95]. For amplification of the *TBKBP1* DMR and subsequent pyrosequencing, we used the following primers:

'forward' 5'-TATTTAAGTTTGGGTGATAGAGTAAGAT-3'
'reverse' 5'-Bio-CCCAACCCTCAAAAATATAATATCT-3'
'sequencing 1' 5'-GGTGGTGTATGTTTGTAAT-3'
'sequencing 2' 5'-GGTTGAGGTGAGTTAAGAT-3'

### Quantitative RT-PCR of *TBKBP1*

Total RNA was purified from isolated CD8+ T cell subsets using the RNeasy Mini Kit (Qiagen), spectrometrically quantified (DeNovix), and transcribed into cDNA using Transcriptor First Strand cDNA Synthesis Kit (Roche). TBKBP1-A primers ('forward' 5'-TTGCCCTCATCAC TGCTTAC-3'; 'reverse' 5'-GGGTACTTGATCTCGTAGACTTTG-3'), TBKBP1-B primers ('forward' 5'-CAGGATCTGGCCTCCAAC-3'; 'reverse' 5'-CCCTGTAGGGAACTCAACTC-3') or codon-optimized TBKBP1 primers ('forward' 5'-TCGCTCTCATCACTGCCTAC-3';

'reverse' 5'-GGGTACTTGATTTCATAGACTTTA-3') and SYBR green master mix (Roche) as well as cDNA were used in a qRT-PCR on a LightCycler 480 II (Roche) or QuantStudio 3 (Thermo Fisher Scientific). The data were analysed with LightCycler 96 SW 1.1 (Roche) or Design & Analysis Software v2.7.0 (Thermo Fisher Scientific). All procedures were performed according to the manufacturers' recommendations.

### Retroviral transduction

For retroviral overexpression of TBKBP1, an expression cassette consisting of codon-optimized *TBKBP1* cDNA, porcine teschovirus-1 2A element (P2A), and mCherry cDNA was inserted into the pMP71 plasmid by gene synthesis (TWIST Bioscience). pMP71 containing only mCherry cDNA served as empty vector (EV) control. For the ARMATA, cells were not only retrovirally transduced with the TBKBP1 overexpression construct (or the EV control), but also with a pMP71-based plasmid coding for mTCR 5–2, a previously reported CMV-specific, HLA-A*02:01-restricted high-avidity TCR recognising the CMVpp65-derived peptide NLVPMVATV and containing a murine constant region (mTCR) for the isolation of successfully transduced cells [70].

Virus transduction of human PBMCs was performed as previously described [96]. In brief, the virus-packaging RD114 cell line was seeded in a 6-well plate at a density of 1.5 x 10$^6$ cells/well 1 day prior to transfection. The next day, pMP71 vectors were transfected using calcium phosphate precipitation. Thereto, complete DMEM medium (Gibco) supplemented with 10% FBS (Gibco), 1% sodium pyruvate (Biochrom), and 1% HEPES (Sigma-Aldrich) was replaced 1 hour before transfection with complete medium containing 25 μM chloroquine diphosphate salt (Sigma-Aldrich). The plasmid-containing buffer was prepared by adding 18 μg of vector DNA and 15 μl of 2.5 M CaCl$_2$ (Applichem) to a final volume of 150 μl with water, which was then mixed with transfection buffer containing 280 mM sodium chloride, 42 mM HEPES, 3.5 mM disodium hydrogen phosphate, and 10 mM potassium chloride at 1:1 ratio by vortexing and subsequently incubated for 30 minutes at room temperature. The mixture was slowly added to RD114 cells (150 μl/well) in a dropwise manner and incubated for 15 hours at 37˚C with 5% CO$_2$, followed by medium replacement with 3 ml complete DMEM medium. The supernatant containing retroviral particles was harvested 3 days later. For transduction, PBMCs were stimulated for 48 hours with plate-bound anti-human CD3 and anti-human CD28 (both 1 μg/ml). Additionally, retronectin-treated plates were prepared by coating with 12.5 μg/ml retronectin (TaKaRa) in PBS (Gibco) at 4˚C overnight in a non-treated 24-well plate. Unbound retronectin was aspirated and plates were blocked with 2% BSA (Sigma-Aldrich) in PBS for 20 minutes at 37˚C followed by washing twice with PBS. Supernatant containing retroviral particles was added to retronectin-coated plates followed by centrifugation at 2,000xg for 2 hours at 32˚C. Following centrifugation, virus supernatant was aspirated and preactivated PBMCs (0.5–0.8 x 10$^6$/well) in TexMACS medium supplemented with 100 IU/ml IL-2 and 6 ng/ml polybrene (Sigma-Aldrich) were added to each well. After spinoculating the cells onto virus-coated plates (1,000xg for 10 minutes at 32˚C), the cells were incubated at 37˚C with 5% CO$_2$ for 24 hours. After 24 hours, transduced human PBMCs were washed and resuspended in TexMACS medium supplemented with IL-2. Flow cytometry analysis was performed 5 days after spinoculation to determine transfection efficiency. Successfully transduced CD8$^+$mCherry$^+$ T cells (TBKBP1 overexpression or EV control) or CD8$^+$mTCR$^+$mCherry$^+$ T cells (simultaneous overexpression of mTCR 5–2 and TBKBP1 or EV control) were sorted using flow cytometry and used for subsequent experiments.

### Cell stimulation and immunoblotting

For sodium dodecyl sulphate polyacrylamide gel electrophoresis (SDS-PAGE) and subsequent immunoblotting experiments, TBKBP1-overexpressing CD8$^+$ T cells and control cells

transduced with EV as well as *ex vivo* isolated CD8$^+$ T$_N$, T$_{CM}$, T$_{EM}$ and T$_{EMRA}$ cells were generated and sorted as described above. Next, cells were serum-starved overnight and subsequently incubated with soluble 2 μg/ml anti-CD3 and anti-CD28 mAbs for 40 minutes on ice. Antibody-coated cells were then cross-linked with 4 μg/ml AffiniPure goat-anti-mouse IgG+IgM (H+L; Jackson ImmunoResearch) and incubated for 15 minutes at 37°C. Unstimulated controls were kept on ice. To stop the stimulation, excess ice-cold PBS was added and the cells were centrifuged at 4°C. Pellets were snap-frozen and stored at -80°C.

For preparing cell lysates from TBKBP1-overexpressing CD8$^+$ T cells and control cells transduced with EV for SDS-PAGE and immunoblotting, cell pellets were resuspended in 40 μl of 2x SDS loading buffer (0.125 M Tris-HCl, pH 6.8, 4% SDS, 20% glycerol, 0.02% bromphenol blue, 10% β-mercaptoethanol), heated at 95°C for 5 minutes, and cell debris was pelleted with 14,000xg at room temperature (RT) for 5 minutes. Per lane, 20 μl lysate was analysed on 10% polyacrylamide gels containing 0.1% SDS in Tris-glycine running buffer with 0.1% SDS at room temperature.

For preparing cell lysates from *ex vivo* isolated CD8$^+$ T cell subsets for SDS-PAGE and immunoblotting, approximately 1 x 10$^6$ cells were resuspended in 20 μl lysis buffer (20 mM Tris-base, 100 mM NaCl, 1 mM EDTA, 1% Triton X-100) supplemented with protease and phosphatase inhibitors (both from Roche). Cells were lysed for 1h at 4°C and cell debris was precipitated at 14,000xg at 4°C for 5 minutes. Protein concentration was assessed colorimetrically via the bicinchoninic acid (BCA) protein assay (Takara), adjusted protein amounts were mixed with 4x SDS loading buffer (0.25 M Tris-HCl, pH 6.8, 8% SDS, 40% glycerol, 0.04% bromphenol blue, 10% β-mercaptoethanol), lysates were heated at 95°C for 10 minutes, and loaded on 7.5% polyacrylamide gels containing 0.1% SDS in Tris-glycine running buffer with 0.1% SDS at room temperature.

Proteins were blotted on polyvinylidene difluoride membranes (Amersham Hybond P 0.2 μm) in a wet blot system in Tris-glycine transfer buffer containing 0.05% SDS and 20% methanol for 1 hour at 350 mA at 4°C. Membranes were subsequently blocked in 5% BSA in TBS with 0.1% Tween 20 for 1 hour at room temperature and incubated with anti-phospho-TBK1/NAK (Ser 172; Cell Signaling Technology), anti-TBK1/NAK (D1B4; Cell Signaling Technology) anti-SINTBAD (TBKBP1; D1A5; Cell Signaling Technology), anti-GAPDH (14C10; Cell Signaling Technology) or anti-α-Tubulin (Sigma-Aldrich) primary antibodies at 4°C overnight. Membranes were washed and incubation with HRP-coupled anti-rabbit or anti-mouse secondary antibodies (both from Dianova) was carried out for 1 hour at room temperature. Membranes were washed and developed with the Pierce ECL or SuperSignal substrate (Thermo Fisher Scientific) and a ChemoStar ECL Imager (INTAS). Quantitative analysis was performed with ImageJ software (version 1.52g, National Institute of Health). When necessary, membranes were stripped with Restore PLUS Western Blot Stripping Buffer (ThermoFisher Scientific) for 15 minutes at room temperature, before being washed, blocked, incubated with primary and secondary antibodies, imaged, and analysed as described above.

## PKCθ inhibition

PKCθ inhibition experiments were carried out with EV-transduced or TBKBP1-overexpressing CD8$^+$ T cells generated as described above. To assess the impact of PKCθ inhibition on TBK1 phosphorylation, sorted CD8$^+$mCherry$^+$ T cells were serum-starved overnight and subsequently preincubated for 1 hour with the PKCθ inhibitor (10 μM HY-112681, Hoelzel Biotech). Next, cells were restimulated for 15 minutes and further processed for SDS-PAGE and immunoblotting as described above. To assess the impact of PKCθ inhibition on the functional properties of CD8$^+$ T cells, sorted CD8$^+$mCherry$^+$ T cells were preincubated for 1 hour with

the PKCθ inhibitor as described above. DMSO-treated cells were used as control. Next, cells were stimulated for 2 hours using T cell TransACT (Miltenyi Biotech) as recommended by the manufacturer. Afterwards, Brefeldin A (10 μg/ml, Sigma-Aldrich) was added and cells were cultivated for additional 2 hours. At the end of the culture, cells were harvested and stained for flow cytometry as described above.

### Assay for rapid measurement of antiviral T cell activity

For ARMATA [68], CD8[+] T cells simultaneously overexpressing mTCR 5–2 and TBKBP1 (or the EV control) were generated and sorted as described above. One day prior to infection, MRC-5 cells were seeded in 96-well plates at a density of 20,000 cells/well and cultured in Eagle's minimum essential medium with 10% FCS, 1 mM sodium pyruvate, and 1 mM gluta-mine. The following day, the MRC-5 cells were infected with a reporter CMV (TB40/BAC4 HCMV[3F]) that was generated on the background of the clinical isolate TB40-E [69]. Briefly, after aspiration of the medium, the reporter CMV was added to the MRC-5 cells at a MOI of 0.1 and centrifuged at 700xg for 10 minutes at room temperature to enhance the infection and reduce infection heterogeneity. After centrifugation, the medium was immediately aspirated and the cells were washed with PBS. After washing, sorted mTCR[+]mCherry[+] CD8[+] T cells were added to the infected MRC-5 cells in a final volume of 200 μl medium at effector:target (E:T) ratios of 0.5:1 and 1:1. Coculture medium was composed of CTS OpTmizer T-cell expansion basal medium (ThermoFischer Scientific) supplemented with CTS Immune Cell SR (ThermoFischer Scientific) and 300 IU/ml IL-2. To analyse the virus reduction capacity of the T cells, the mNeonGreen signals from infected MRC-5 cells were monitored for 1 week using the Incucyte Live-Cell analysis system (Sartorius). Additionally, the confluence of cells was determined by the phase detector and images were analysed by the Incucyte Analysis Software. Data were exported to Prism 9.4.0 for graphical representation.

### Cytokine multiplex assay

For measuring cytokines, 50 μl culture supernatants were harvested after 36 hours of the co-culture from the ARMATA. Supernatants were centrifuged at 500xg for 15 minutes at 4˚C and kept at -80˚C. For cytokine measurement, 50 μl thawed supernatant was diluted 1:1 with supplied assay buffer. Cytokines were measured using the MILLIPLEX Human CD8[+] T Cell Magnetic Bead Panel Premixed 17 Plex-Immunology Multiplex Assay (Millipore) according to the manufacturer's recommendations and analysis was done using the Bio-Plex Manager 6.1 software based on the standard curve.

### Statistical analysis

Prism 9.4.0 software was utilized for statistical analysis. The normal distribution of data points was checked using the Shapiro-Wilk normality test. A paired two-tailed parametric student's $t$ test was conducted on samples that passed the normality test, and a non-parametric Wilcoxon matched-pairs signed rank test was performed on samples that failed in normality test. All data are presented as mean or mean ± SD, and $p$-values $\leq$ 0.05 deemed significant (* $p \leq 0.05$; ** $p \leq 0.01$; *** $p \leq 0.001$; **** $p \leq 0.0001$; ns; not significant).

### Supporting information

**S1 Fig. Phenotypic characterisation of T(CMV) cells.** PBMCs from healthy CMV-seroposi-tive donors were pre-enriched for CD8[+] T cells and subsequently stimulated with the CMVpp65 overlapping peptide pool. Next, IFN-γ-secreting cells were detected using the IFN-

γ Secretion Assay/Detection Kit. The phenotype of IFN-γ-secreting T(CMV) cells was determined using flow cytometry. **(A)** Representative flow cytometry plots show the identification of T(CMV) cells (left) and the phenotypic characterisation via CD45RA and CD62L expression (right). Numbers indicate frequencies in gates or quadrants. **(B)** The bar plot shows the frequencies of $T_N$ (CD45RA$^+$CD62L$^+$), $T_{CM}$ (CD45RA$^-$CD62L$^+$), $T_{EM}$ (CD45RA$^-$CD62L$^-$) and $T_{EMRA}$ cells (CD45RA$^+$CD62L$^-$) within T(CMV) cells from 4 donors. Black dots indicate frequencies from individual donors and grey bar mean values with SD.
(PDF)

**S2 Fig. Sorting of CD8$^+$ T cell subsets for WGBS.** PBMCs from healthy CMV-seropositive donors were pre-enriched for CD8$^+$ T cells and stimulated with CMVpp65 overlapping peptide pool to detect IFN-γ-secreting T(CMV) cells using IFN-γ Secretion Assay/Detection Kit. Representative flow cytometric plots show the gating strategy for sorting of $T_N$ (CD3$^+$CD4$^-$CD8$^+$CCR7$^+$CD28$^{int}$KLRG1$^-$CX$_3$CR1$^-$CD45RA$^{high}$) (orange gate), $T_{mem}$ (CD3$^+$CD4$^-$CD8$^+$CCR7$^-$CD28$^{high}$CD27$^+$CD45RA$^-$) (light blue gate), and T(CMV) cells (CD3$^+$CD4$^-$CD8$^+$IFN-γ$^+$) (red gate) from 5 donors.
(PDF)

**S3 Fig. Correlation between methylation level of *TBKBP1* DMR and *TBKBP1* expression in CD8$^+$ T cells from CMV-seronegative donors. (A)** Mean methylation level of *TBKBP1* DMR in sorted CD8$^+$ T cell subsets $T_N$, $T_{SCM}$, $T_{CM}$, $T_{EM}$, and $T_{EMRA}$ cells from five CMV-seronegative donors are shown. **(B)** *TBKBP1* expression in indicated CD8$^+$ T cell subsets from five CMV-seronegative donors were analysed by RT-PCR. **(C)** Correlation analysis visualize the relation between methylation status of *TBKBP1* DMR (x-axis) and associated *TBKBP1* gene expression (y-axis) including the linear regression line and the Pearson correlation coefficient (r).
(PDF)

**S4 Fig. Alternative transcripts generated from the *TBKBP1* gene locus. (A)** Overview of the exon/intron structure of putative transcripts arising from the *TBKBP1* gene locus on chromosome 17 (source: Ensembl GRCh38.p14). White boxes show untranslated exons, whereas black boxes represent translated exons. Ensembl transcript code, amino acid (aa) length of the protein, primer pairs for RT- PCR and the position of the *TBKBP1* DMR are indicated. **(B)** Ratio of normalized RT-PCR signals from amplificated TBKBP1-A and TBKPB1-B using RNA of $T_{CM}$, $T_{EM}$ and $T_{EMRA}$ cells. For statistical analyses, a paired two-tailed student's t test was conducted with *, p $\leq$ 0.05.
(PDF)

**S5 Fig. Phenotypic characterisation of CD8$^+$ $T_N$ and $T_{EMRA}$ cells during long-term cultivation.** CD8$^+$ $T_N$ and $T_{EMRA}$ cells were sorted from healthy CMV-seropositive donors and cultured up to 30 days with repetitive restimulations using plate-bound anti-human CD3 and anti-human CD28 antibodies. Every 5 days, cells were harvested, washed and an aliquot was collected to determine the expression of CD8, CCR7, CD45RA, CD62L, CD95, and CD28 by flow cytometry. Representative flow cytometry plots from 5 independent cultures are depicted.
(PDF)

**S6 Fig. Validation of retroviral TBKBP1 overexpression in CD8$^+$ T cells. (A)** Workflow describing the steps involved in sample processing for immunoblotting experiments. Briefly, after 48 hours of activation, PBMCs obtained from CMV-seronegative donors were transduced with pMP71-based vectors. From both empty vector (EV)- and TBKBP1-transduced samples, CD8$^+$mCherry$^+$ cells were sorted according to the depicted gating strategy and

restimulated with anti-human CD3 and anti-human CD28 antibodies for 15 minutes followed by cross-linking with goat anti-mouse IgG, while keeping unstimulated cells as controls. Both unstimulated and restimulated cells from EV and TBKBP1 samples were subjected to immunoblotting to determine the expression levels of TBKBP1, TBK1, pTBK1, and GAPDH. **(B)** Representative flow cytometry plots show gating strategy (left) for the sorting of CD8⁺mCherry⁺ T cells (top: EV control; bottom: TBKBP1 overexpression) and post-sort purity (right) from one out of four donors. **(C)** Bar plots showing the mRNA expression of *TBKBP1* in sorted CD8⁺mCherry⁺ T cells from both TBKBP1-overexpressing samples and EV-transduced controls relative to *RSP9* mRNA expression.
(PDF)

**S7 Fig. Effect of PKCθ inhibition in CD8⁺ T cells on phosphorylation of TBK1.** TBKBP1-overexpressing CD8⁺ T cells and EV-transduced controls were generated, and successfully transduced CD8⁺mCherry⁺ T cells were sorted by flow cytometry, serum-starved overnight, preincubated with the PKCθ inhibitor or DMSO as control, and subsequently short-term stimulated. Samples from EV-transduced or TBKBP1-overexpressing CD8⁺ T cells were subjected to immunoblotting to determine the expression of TBKBP1, pTBK1, and total TBK1. The analysis of α-Tubulin expression served as loading control. Data from two independent donors are depicted.
(PDF)

**S8 Fig. Gating strategy for sorting of TBKBP1-overexpressing CD8⁺ T cells and EV-transduced controls.** PBMCs isolated from healthy CMV-seronegative donors were stimulated with plate-bound anti-human CD3 and anti-human CD28 antibodies and subsequently co-transduced with mTCR and TBKBP1- or EV-mCherry plasmids. Successfully transduced CD8⁺mTCR⁺mCherry⁺ T cells were sorted using flow cytometry. Representative flow cytometry plots from 3 independent donors show the gating strategy for sorting of CD8⁺mTCR⁺mCherry⁺ T cells and post-sort purity.
(PDF)

**S9 Fig. Quantification of cytokines from the ARMATA.** PBMCs isolated from healthy CMV-seronegative donors were stimulated with plate-bound anti-human CD3 and anti-human CD28 antibodies and subsequently co-transduced with mTCR and TBKBP1- or EV-mCherry plasmids. Successfully transduced CD8⁺mTCR⁺mCherry⁺ T cells were sorted from both TBKBP1-overexpressing samples and EV-transduced controls using flow cytometry and co-cultured with CMV-infected MRC-5 cells followed by the ARMATA. 36 hours after infection, culture supernatants were harvested and cytokine profiles determined from cultures of CMV-infected MRC-5 cells without the addition of CD8⁺ T cells (w/o T cells), in the presence of EV-transduced mTCR 5–2⁺ CD8⁺ T cells (EV) or in the presence of TBKBP1-overexpressing mTCR 5–2⁺ CD8⁺ T cells (TBKBP1) with E:T ratios of 0.5:1 (left) and 1:1 (right) for indicated cytokines. Data for IFN-γ, Granzyme A, Granzyme B and Perforin from the E:T ratio of 1:1 are shown in Fig 7. Data from 3 independent experiments with 2 technical replicates each are shown. Black dots indicate frequencies from individual donors and grey bar mean values with SD. For statistical analyses, a paired two-tailed student's *t* test (leaving out "w/o T cell" group) was conducted with *, $p \leq 0.05$ and **, $p \leq 0.01$.
(PDF)

**S1 Table. Identified DMRs including methylation values from all pair-wise comparisons.**
(XLSX)

## Acknowledgments

We thank Lothar Gröbe for carrying out flow cytometry–based cell sorting. We acknowledge the Genome Analytics Facility at the Helmholtz Centre for Infection Research. We appreciate Kerstin Beushausen and Jana Keil from Hannover Medical School for cytokine measurements. We also acknowledge Jana Reinking and Lothar Jänsch for their assistance with protein quantification and CD3/CD28 cross-linking experiments.

## Author Contributions

**Conceptualization:** Britta Eiz-Vesper, Jochen Huehn.

**Data curation:** Andreas Keller, Christine Falk, Dirk H. Busch, Kilian Schober, Luka Cicin-Sain, Fabian Müller, Melanie M. Brinkmann, Stefan Floess.

**Formal analysis:** Zheng Yu, Varun Sasidharan-Nair, Thalea Buchta, Agnes Bonifacius, Fabian Müller, Britta Eiz-Vesper, Stefan Floess, Jochen Huehn.

**Funding acquisition:** Britta Eiz-Vesper, Jochen Huehn.

**Investigation:** Zheng Yu, Varun Sasidharan-Nair, Thalea Buchta, Agnes Bonifacius, Jochen Huehn.

**Methodology:** Zheng Yu, Varun Sasidharan-Nair, Thalea Buchta, Agnes Bonifacius, Fawad Khan, Beate Pietzsch, Hosein Ahmadi, Michael Beckstette, Jana Niemz, Philipp Hilgendorf, Philip Mausberg, Andreas Keller, Christine Falk, Dirk H. Busch, Kilian Schober, Luka Cicin-Sain, Fabian Müller, Melanie M. Brinkmann, Britta Eiz-Vesper, Stefan Floess, Jochen Huehn.

**Resources:** Andreas Keller, Christine Falk, Dirk H. Busch, Kilian Schober, Luka Cicin-Sain, Fabian Müller, Melanie M. Brinkmann, Britta Eiz-Vesper.

**Software:** Zheng Yu, Varun Sasidharan-Nair.

**Supervision:** Jochen Huehn.

**Validation:** Zheng Yu, Varun Sasidharan-Nair, Thalea Buchta, Fawad Khan, Beate Pietzsch, Hosein Ahmadi, Michael Beckstette, Jana Niemz, Philipp Hilgendorf, Philip Mausberg, Andreas Keller, Christine Falk, Dirk H. Busch, Kilian Schober, Luka Cicin-Sain, Fabian Müller, Melanie M. Brinkmann, Stefan Floess, Jochen Huehn.

**Visualization:** Zheng Yu, Varun Sasidharan-Nair, Thalea Buchta, Agnes Bonifacius, Andreas Keller, Christine Falk, Dirk H. Busch, Kilian Schober, Luka Cicin-Sain, Fabian Müller, Melanie M. Brinkmann, Stefan Floess.

**Writing – original draft:** Zheng Yu, Varun Sasidharan-Nair, Kilian Schober.

**Writing – review & editing:** Thalea Buchta, Agnes Bonifacius, Luka Cicin-Sain, Melanie M. Brinkmann, Stefan Floess, Jochen Huehn.

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
