## [Decision Letter · Decision Letter 0]

18 Mar 2024

Dear Dr. Huehn,

Thank you very much for submitting your manuscript "DNA methylation profiling identifies TBKBP1 as potent amplifier of cytotoxic activity in CMV-specific human CD8+ T cells" for consideration at PLOS Pathogens. As with all papers reviewed by the journal, your manuscript was reviewed by members of the editorial board and by several independent reviewers. In light of the reviews (below this email), we would like to invite the resubmission of a significantly-revised version that takes into account the reviewers' comments.

Dr. Huehn,

Thank you for your submission. I do fully apologize for the extended time needed for review. It was particularly difficult to secure three reviewers but in the end we did find three experts in this area and were able to secure a comprehensive review. All three were in agreement with the importance of the study but feel that some additional experimentation and clarification is needed before it is acceptable. With that in mind I would encourage you to address the concerns of the reviewers and resubmit the manuscript for consideration. Again, apologies on the long review process.

Cheers,

Eain Murphy

We cannot make any decision about publication until we have seen the revised manuscript and your response to the reviewers' comments. Your revised manuscript is also likely to be sent to reviewers for further evaluation.

Sincerely,

Eain A Murphy, Ph.D.

Academic Editor

PLOS Pathogens

Blossom Damania

Section Editor

PLOS Pathogens

Michael Malim

Editor-in-Chief

PLOS Pathogens

orcid.org/0000-0002-7699-2064

Dr. Huehn,

Thank you for your submission. I do fully apologize for the extended time needed for review. It was particularly difficult to secure three reviewers but in the end we did find three experts in this area and were able to secure a comprehensive review. All three were in agreement with the importance of the study but feel that some additional experimentation and clarification is needed before it is acceptable. With that in mind I would encourage you to address the concerns of the reviewers and resubmit the manuscript for consideration. Again, apologies on the long review process.

Cheers,

Eain Murphy

Reviewer's Responses to Questions

**Part I - Summary**

Reviewer #1: Yu and colleagues have investigated the methylation status of CMV-specific T cells in humans. They unexpectedly uncovered a demethylation of the gene TBKBP1 in T cells selectively in CMV-specific T cells, most of which are TEMRA, compared to bulk memory T cells. The authors demonstrate that demethylation correlates with expression of TBKBP1. Additionally, using an overexpression system, the investigators suggest that elevated levels of TBKBP1 leads to enhanced phosphorylation of TBK1 and improved T cell killing of CMV-infected cells. In general, the manuscript is well written and the data that were presented are clear. However, the complete reliance on the overexpression system for drawing conclusions about the function of TBKBP1 in T cells is a major issue and leaves the overall story suggestive rather than conclusive.

Reviewer #2: An interesting study that assesses DNA methylation patterns in CMV-specific T cells and identifies a protein TBKBP1 that enhances cytotoxicity. Uncovers a range of novel findings regarding virus-specific differentiation and a valuable dataset for other researchers to interrogate.

Reviewer #3: The study by Yu et al aims to find a DNA methylation signature of virus-specific (cytomegalovirus in this case) CD8-pos T cells and so they initially investigate the genome-wide DNA methylation profiles of T cell populations, namely bulk/total, memory and effector (IFNγ-pos) T cells. Overall, they find that memory and effector genome-wide profiles are starkly different from bulk, although some individual differentially methylation regions (DMRs) also occur between memory and effector populations. One such site occurs within the TBKBP1 gene, which has previously not been investigated in relation to T cell differentiation. More specific DNA methylation analysis (pyrosequencing) of this site across further T cell populations found a decreasing association of DNA methylation as cells become more differentiated/specialized, which is inversely related to expression/transcript level of TBKBP1. Moreover, the DNA methylation level at this site remains stable across 30 days of cell proliferation. Since another group had previously found in epithelial cells that TBKBP1 aids the phosphorylation of TBK1, via CARD10 and PKCθ, and subsequent activation of downstream gene expression, Yu et al confirmed that over-expression of TBKBP1 in and stimulation of T cells is associated with phosphorylation of TBK1. Furthermore, over-expression of TBKBP1 in T cells within a virus reduction assay (akin to a virus dissemination assay or cell killing assay) was associated with an increased ability of these cells to control CMV infection.

Strengths: the study is well performed, presented and written. The finding that TBKBP1 appears to promote cytotoxic CD8-pos T cells activity is novel, although the potential for clinical exploitation remains unclear.

Weaknesses: although the initial DMR findings are very striking, this analysis was only performed on cells from 5 (or in the case of Tn, 4) donors. Also, the significance of this finding is limited due to the final conclusions being drawn largely based on correlative data. Therefore, further control experiments must be conducted to confirm the authors conclusions.

**Part II – Major Issues: Key Experiments Required for Acceptance**

Reviewer #1: 1) Key missing piece of data: enhanced phosphorylation of TBK1 in the TEMRA cells or CMV-specific TEMRA in the absence of TBKBP1 overexpression. Overexpression experiments can lead to non-physiological outcomes and it remains possible that only artificially very high levels of TBKBP1 have an effect on TBK1 and T cell function. If so, the demthylation of TBKBP1 may not lead to any change in T cell function, but have some other effects.

2) Additionally, the relative amount of TBKBP1 and TBK1 in the overexpression system vs the unmanipulated TEMRA cells should be shown. Is the expression too low to detect by Western blot without overexpression?

3) Ideally, the investigators should also show that knockdown of TBKBP1 in TEMRA cells would reduce pTBK1 and T cell efficacy. Even if levels of TBKBP1 are too low to detect by Western blot, a change in function should be evident if their conclusions are not an artifact of overexpression.

4) For the methylation patterns, CMV-specific TEMRA were compared to bulk memory and naïve CD8 T cells. In general, this is reasonable but it leaves out some information. It wasn’t clear to me whether the results were specific to TEMRA or CMV-specific T cells or CMV-specific TEMRA. Are the same patterns true the TBKBP1 levels and methylation are compared to CMV-specific Tmem, or bulk TEMRA cells regardless of specificity or TEMRA cells in CMV-negative donors? I know there are smaller numbers of CMV-specific Tmem and TEMRA cells in CMV-negative donors, but there should be enough for qPCR of TBKBP1 levels at least, if not some analyses of methylation patterns.

Reviewer #2: Nil

Reviewer #3: Major issues to address

Fig. 3 – all data here correlates DNA methylation levels with mRNA/transcripts levels. Transcript levels are determined by qPCR with primers spanning exons 1-2, if correct (although please see note in Minor issues regarding primers also). However, the DMR is further down the gene beyond further intron/exon breaks. DNA methylation is also known to govern splicing events, which is also linked to differentiation of cells and changing transcript profiles. Therefore, the authors must show the total profile of possible transcripts from this gene (i.e. regular PCR minimally or RNA-seq) across their differentiating cell populations and also show protein expression levels of TBKBP1, which of course is the presumed effector of their final phenotype.

Fig. 4 – since the authors hypothesize in Fig 4A that PKCθ is the kinase responsible for phosphorylation of TBK1 during their experiment, to test this fully they must test this through use of a highly selective PKCθ inhibitor (e.g. HY-112681). Moreover, immunoprecipitation experiments would also be useful to show that TBKBP1/TBK1 and/or TBKBP1/CARD10 are interacting under the conditions of their experiment to prove direct effects.

Fig. 5 – the data here are compelling, however further inhibitor use during experiments should help to confirm that the pathway the authors believe is activated is the actual effector, i.e. use of highly selective PKCθ (e.g. HY-112681) and/or TBK1 (e.g. GSK8612) inhibitors, which should eliminate the difference in killing between EV and TBKBP1 over-expressing cells. This type of experiment presumably could also be conducted with CD8-pos T cells from sero-positive donors and CMV infected target cells to show a decrease in the natural ability of these cells to kill.

**Part III – Minor Issues: Editorial and Data Presentation Modifications**

Reviewer #1: While the trends are obvious in Figure 5C, the differences appear to be very subtle and the power of the analyses appears weak with the possible exception of IFNg expression. I believe that the samples derive from individual donors transfected with either the EV or the TBKBP1. It might help if the paired samples were indicated in the figure as in 5B.

Reviewer #2: This is an interesting and well performed study.

I do not see the need for further experimental work.

One point is that the CMV-specific T cells are (1) recently activated and (2) EMRA phenotype. These are compared against CD27+CD28+ early differentiated effector T cells without recent stimulation. Whilst later experiments show that TBKTB1 methylation is not altered by cell proliferation, and pyrosequencing of EM vs EMRA is undertaken, I would mention this limitation in the Discussion section.

The work of the Peterson group in this area should be mentioned and discussed - e.g. PMID: 38370415; PMID: 35397197.

There is also some work on genetic variants of TBKBP1 with HHV7 infection (PMID: 33109261)

Reviewer #3: Minor issues to address

Fig. 2 – do all the genes shown with DMRs here have changes to transcript level correlating with DNA methylation level consistent with TBKBP1 data, or is this restricted to TBKBP1? Is qPCR data available for other genes as per Fig. 3B/C. This of course would determine the importance of DMR profiling across T cell populations, since only one gene has been focused up beyond this point.

Fig. 5A – brightfield images would be useful to show the absence of further cells. Although the T cells are presumably much smaller, can their presence be shown with the red/mCherry channel here?

Discussion

Is there any known correlation between TBKBP1 mutations and natural susceptibility to virus/herpesvirus infections?

Reduced DNA methylation at the DMR in TBKBP1 is clear during differentiation, however some discussion must be made regarding why this site in particular is important – further areas in the same intron appear also de-methylated vs the two other cell populations.

Also, is there an enhancer element at the TBKBP1 DMR, since the methylation is not lost near the promoter as is highlighted as important by the authors themselves (lines 238-240)?

M&Ms – the primers used for qPCR have a number of mismatches with the available human TBKBP1 gene sequence available from NCBI. Is there a particular reason for this?

Supplement

S2 Fig – cell type is denoted by box color here – does this mean that all black boxes are TN (naïve T cells)?

Fig S6 – unlike killing of target cells, which becomes more statistically significant with an E:T ratio of 1 (vs 0.5; Fig 5B), statistical significance of changed expression of cytokines here appears generally to decrease with a greater E:T. Can this be explained?

S6 Fig. legend – it must be mentioned that some data presented here is also already presented in Fig. 5.

PLOS authors have the option to publish the peer review history of their article (what does this mean?). If published, this will include your full peer review and any attached files.

Reviewer #1: No

Reviewer #2: **Yes: **Paul Moss

Reviewer #3: No
---

## [Editor Report · Decision Letter 1]

11 Sep 2024

Dear Dr. Huehn,

We are pleased to inform you that your manuscript 'DNA methylation profiling identifies TBKBP1 as potent amplifier of cytotoxic activity in CMV-specific human CD8+ T cells' has been provisionally accepted for publication in PLOS Pathogens.

Best regards,

Eain A Murphy, Ph.D.

Academic Editor

PLOS Pathogens

Blossom Damania

Section Editor

PLOS Pathogens

Michael Malim

Editor-in-Chief

PLOS Pathogens

orcid.org/0000-0002-7699-2064

Dr. Huehn,

Thank you for resubmitting your manuscript. Your resubmission has been reviewed by the editorial staff and we have come to the conclusion that your response to reviewers and modifications of the manuscript have significantly improved the work. It is our opinion that there remain no further concerns about the quality of the work and have come to the decision of ACCEPT without additional review. Congratulations, this is a nice piece of work and you and your colleagues should be proud.

Cheers,

Eain Murphy
---

## [Editor Report · Acceptance letter]

20 Sep 2024

Dear Dr. Huehn,

We are delighted to inform you that your manuscript, "DNA methylation profiling identifies TBKBP1 as potent amplifier of cytotoxic activity in CMV-specific human CD8+ T cells," has been formally accepted for publication in PLOS Pathogens.

Best regards,

Michael Malim

Editor-in-Chief

PLOS Pathogens

orcid.org/0000-0002-7699-2064